



# The importance of accurate glacier albedo for estimates of surface mass balance on Vatnajökull: Evaluating the surface energy budget in a Regional Climate Model with automatic weather station observations

Louise Steffensen Schmidt[1], Guðfinna Aðalgeirsdóttir[1], Sverrir Guðmundsson[1,2], Peter L. Langen[3], Finnur Pálsson[1], Ruth Mottram[3], Simon Gascoin[4], and Helgi Björnsson[1]

[1]University of Iceland, Institute of Earth Sciences, Reykjavik, Iceland
[2]Keilir Institute of Technology, Reykjanesbær, Iceland
[3]Danish Meteorological Institute, Copenhagen, Denmark
[4]Centre d'Etudes Spatiales de la Biosphère, Université de Toulouse, CNES/CNRS/IRD/UPS, Toulouse, France

*Correspondence to:* L.S. Schmidt (lss7@hi.is)

**Abstract.** A simulation of the surface climate of Vatnajökull ice cap, Iceland, made with the Regional Climate Model HIRHAM5 for the period 1980-2014, is used to estimate the evolution of the glacier mass balance. A new snow albedo parametrization is used for the simulation that describes the albedo with an exponential decay with time and is surface temperature dependant. The albedo scheme utilizes a new background map of the ice albedo created from observed MODIS data. The simulation is

evaluated against observed daily values of weather parameters from five Automatic Weather Stations (AWSs) from 2001-2014, as well as in situ mass balance measurements from 1994-2014. The model simulates the observed parameters well at the station sites, albeit with a general underestimation of the net radiation. This is due to an underestimation of the incoming radiation and a general overestimation of the albedo. The average modelled albedo is overestimated in the ablation zone, which we attribute to an overestimation of the thickness of the snow layer and not taking dirt and volcanic ash deposition during dust storms and

volcanic eruptions into account. A comparison with the specific summer, winter, and net mass balance for all of Vatnajökull from 1994-2014 shows a good overall fit during the summer, with the model underestimating the balance by only 0.04 m w.eq. on average, but a too large winter balance due to an overestimation of the precipitation at the highest areas of the ice cap. The average overestimation of the winter balance is 0.5 m w.eq., but a simple correction of the accumulation at the highest points of the glacier reduces this to 0.15 m w.eq. The model captures the evolution of the specific mass balance well, for example

capturing a shift in the balance in the mid-1990s, which gives us confidence in the results for the entire model run. The model is therefore used to provide an estimate of the evolution of the specific surface mass balance of Vatnajökull from 1981, and we show the importance of bare glacier ice albedo to modelled mass balance and that processes not currently accounted for in RCMs, such as dust storms, are an important source of uncertainty in estimates of snow melt rate.

**Keywords:** glacier; albedo; energy balance; HIRHAM5



# 1 Introduction

Worldwide, glaciers and ice caps are losing mass at increasing rates as a response to climate change (e.g. Vaughan et al., 2013). Major changes in the dimensions of glaciers are expected to affect the sea level and climate throughout the world, and it is therefore important to describe and understand the glacier climate. Glacier retreat and mass loss at significantly increasing rates

are also observed for Icelandic glaciers (Björnsson et al., 2013), which could potentially contribute to sea level rise by 1 cm (Björnsson and Pálsson, 2008; Björnsson et al., 2013). Numerical high-resolution Regional Climate Models (RCMs), such as MAR (Gallée and Schayes, 1994), RACMO2 (Meijgaard et al., 2008), or HIRHAM5 (Christensen et al., 2006), are valuable tools for estimating the meteorological parameters and mass balance variability at the surface of glaciers. However, in order to have confidence in the result of future model projections, or model reconstructions of past climate, it is important to evaluate

how well they simulate the present climate.

Evaluation of RCMs is important, not only because it reveals possible biases in the model, but also because it could yield recommendations for model improvements. Much work has gone into evaluating RCMs over Greenland (e.g. Box and Rinke, 2003; Noël et al., 2015; Rae et al., 2012; Langen et al., 2016) and Antarctica (e.g. Lenaerts and Van Den Broeke, 2012; Agosta et al., 2015), but less effort has gone into evaluating them over Iceland (e.g. Ágústsson et al., 2013; Nawri, 2014).

However, since a long term meteorological monitoring programme has been conducted on Icelandic glaciers since the 1991-92 glaciological year (e.g. Björnsson et al., 1998), Icelandic glaciers make an excellent evaluation site for the meteorological and mass balance components simulated by a RCM. The runoff from Vatnajökull ice cap is economically important to hydropower production in Iceland and the present and future mass balance is thus of keen interest. Furthermore, the good records over a relatively small area, compared to Greenland, offer a good opportunity to evaluate the spatial and temporal variability of

the HIRHAM5 model. As the albedo may be significantly different in Iceland than that of Greenland or Antarctica, e.g. due to frequent dust storms and occasional volcanic eruptions, model evaluations over Iceland can provide important insight into the effect of albedo changes on the glacier energy balance on a regional scale.

The RCM used in this study is HIRHAM5, which is a state of the art, high-resolution RCM that has been well validated over Greenland (e.g. Box and Rinke, 2003; Lucas-Picher et al., 2012; Rae et al., 2012; Langen et al., 2016). Here we present a

HIRHAM5 simulation with an updated albedo scheme which uses a background albedo map in an attempt to take the effects of dust and tephra on the glacier ice in the ablation zone into account. Model simulation results are compared to observations from Automatic Weather Stations (AWS) and in situ mass balance observations, in an effort to improve the performance of the model. The possible physical reasons for any model biases are discussed, and recommendations for corrections are made where possible.

Furthermore, the mass balance of Vatnajökull is reconstructed back to 1981 using the model, keeping in mind the identified model errors.



## 2 Observational data

The primary observational dataset used in this study was collected by AWSs at selected locations on Vatnajökull. Since 1994, 1-13 stations have been operated on the ice cap during the summer months (e.g. Oerlemans et al., 1999; Guðmundsson et al., 2006). The temperature, relative humidity, wind speed, and wind direction at 2 m above the surface have been measured during

the entire period, while the radiation components have been measured since 1996. For this study, data from five AWSs were considered - three on Brúarjökull (B) and two on Tungnaárjökull (T) (see Figure 1). Two stations are situated in the ablation zone (henceforth referred to as the AB stations), one station is situated near the equilibrium line altitude (ELA station), and two stations are in the accumulation zone (AC stations). The average elevation of each station is shown in Table 1. All five stations have been operated on the glacier every year during the period 2001-2014. Observations of 2 m temperature, humidity, wind

speed, and radiative fluxes were used to validate HIRHAM5 over Vatnajökull.

The uncertainties of the AWS observations vary depending on the sensor. The temperature and humidity sensors have an accuracy of 0.2 °C and 2 % for temperature and humidity, respectively, while the accuracy of the wind speed is 0.2 ms$^{-1}$ (Guðmundsson et al., 2009). The radiative fluxes were measured using either Kipp and Zonen CM14, CNR1 or CNR4 sensors that have a maximum manufacturer-reported uncertainty of ±10 % for daily totals (e.g. Kipp and Zonen, 2002). However, the

uncertainty has independently been evaluated to be lower (3-5 %) when used in an ice sheet environment (van den Broeke et al., 2004; Guðmundsson et al., 2009). The turbulent fluxes and surface pressure were not measured at the stations, but were estimated using the methods described in Section 3.1.

In addition to AWS data, in situ mass balance measurements were used to evaluate the simulated surface mass balance (SMB)

at several sites on Vatnajökull. Conventional in situ mass balance measurements have been carried out every glaciological year since 1991-92, with 60 stations measured each year on average. The measurement sites are shown in Figure 1. The uncertainty of the mass balance measurements has been estimated to be ±0.3 m w.eq.

The SMB measurements are conducted at the beginning and end of the accumulation season in order to measure both the winter and summer balance. The winter balance is measured in the beginning of the melt season by drilling down to the pre-

vious summer layer and weighing the snow column. The summer surface is used as the reference level even if some snow accumulation had occurred by the time the summer balance measurements were conducted. The snow thickness on top of the summer surface at the time of the autumn survey has been measured since the 1995. This is needed when comparing with the simulation of snow accumulation.

Observations of the broadband albedo in the shortwave domain (0.3-5.0$\mu$m) from the MODerate Resolution Imaging Spectro-radiometer (MODIS) was used to create a background map of the ice albedo at all glacier gridpoints in HIRHAM5, which was used in the implemented HIRHAM5 albedo scheme. The MODIS estimates of the albedo on Vatnajökull have been shown to be in good agreement with AWS data (Gascoin et al., 2017). The MODIS data were extracted in geographical coordinates (lon/lat) at a resolution of 0.005°, i.e. close to the original MODIS resolution of 500 m. This was done using the MODIS reprojection



tool with the bilinear interpolation method. These MODIS data in lat/lon were then resampled to match the rotated HIRHAM5 lon/lat grid coordinates by bilinear interpolation using Matlab's interpn function (MATLAB, 2015).

## 3  Model description

### 3.1  AWS point models

The turbulent energy fluxes were calculated using a one-level eddy flux model (Björnsson, 1972; Guðmundsson et al., 2009) which uses Monin-Obukhov similarity theory (Monin and Obukhov, 1954) and has implemented different roughness lengths for the vertical profiles of wind, temperature, and water vapour (Andreas, 1987). The model is described in detail in Guðmundsson et al. (2009). Uncertainties of this model for example pertain to the aerodynamic roughness length for momentum $z_0$. The majority of $z_0$ values recorded over melting glacier surfaces vary over two orders of magnitude (between 1 and 10 mm), but

over fresh snow or smooth ice surfaces the roughness length is generally around 0.1 mm (Brock et al., 2006). An order of magnitude increase in $z_0$ can more than double the estimated turbulent fluxes (Brock et al., 2000), so the chosen roughness length parametrization can greatly affect the performance of the model. Generally, a constant value of $z_0$ is prescribed for snow and/or ice surfaces (Brock et al., 2006), which is an oversimplification as the roughness may vary significantly over the ablation season (e.g. Grainger and Lister, 1966).

However, since measurements of the evolution of $z_0$ over the entire measurement period are not available, a constant roughness length of 1 mm was chosen in the calculation of the non-radiative fluxes. Sensitivity tests were conducted to estimate how large an error this choice of roughness length could lead to at the used AWS sites. A roughness length of 0.1 mm would decrease the calculated results by 16-22 %, while using a roughness length of 10 mm would increase the calculated fluxes by 10-19 %, depending on the station. Since the contribution of the turbulent fluxes to the total energy balance is generally low,

this translates into an increase or a decrease in the total energy balance at the stations by a maximum of 7 %.

The surface air pressure at the station is also needed to calculate the turbulent fluxes, but it is not measured at the AWS sites. Instead it is estimated at the relevant elevation $h$ using synoptic observations from meteorological stations operated by the Icelandic Met Office and the following relationship:

$$P(h) = P(h_0) \left( 1 - \frac{0.0065(h - h_0)}{T(h_0)} \right)^{5.25} \tag{1}$$

where $P(h_0)$ and $T(h_0)$ are the air pressure and air temperature, respectively, observed at an elevation $h_0$ (e.g Wallace et al., 2006). This method has previously been applied successfully at various locations on Vatnajökull and Langjökull (e.g. Guðmundsson et al., 2006, 2009).





## 3.2 HIRHAM5

In this study we employed the regional climate model HIRHAM5 (Christensen et al., 2006), which was developed at the Danish
Meteorological Institute. It is a hydrostatic RCM which combines the dynamical core of the HIRLAM7 numerical forecast-
ing model (Eerola, 2006) and physics schemes from the ECHAM5 general circulation model (Roeckner et al., 2003). Model
simulations have been successfully validated over Greenland using AWS and ice core data (e.g. Box and Rinke, 2003; Stendel
et al., 2008; Lucas-Picher et al., 2012; Langen et al., 2015; Rae et al., 2012; Langen et al., 2016)

While the original HIRHAM5, as described in Christensen et al. (2006), used unchanged ECHAM physics, an updated model
version, which includes a dynamic surface scheme that explicitly calculates the surface mass budget on the surface of glaciers
and ice sheets, is used in this study. This new scheme takes melting of snow and bare ice into account and resolves the retention
and refreezing of liquid water in the snow pack (Langen et al., 2015, 2016). In addition, the 5 layer surface scheme in ECHAM
has been expanded to 25 layers.

### 3.2.1   New albedo parametrization

The updated model also features a more sophisticated snow albedo scheme (Nielsen-Englyst, 2015) than that used in the
original HIRHAM5; whereas the previous scheme was purely temperature dependent, the new scheme depends both on the
age of the snow and the surface temperature. The scheme is similar to that used in Oerlemans and Knap (1998), which assumes
that the albedo decays exponentially as it ages, but in this study an additional temperature component is applied. If there is
snow on the surface, the change in the snow albedo from one time step to the next depends on whether the surface is in a cold
($<$-2 $°C$) or wet regime ($\geq -2\,°C$). In the cold regime, the surface temperature is too low for any melting to occur, while in the
wet regime the temperature in the surface layer is high enough for the surface to be melting. The snow albedo changes over a
timestep, $\delta t$, as

$$\alpha_{snow}^t = (\alpha_{snow}^{t-1} - \alpha_{mx}) \cdot e^{-\delta t/\tau_x} + \alpha_{mx} \qquad (2)$$

where $\alpha_{mx}$ is the minimum snow albedo value that can be reached from ageing of the snow and $\tau_x$ is a timescale which
determines how fast the albedo reaches its minimum value. These two variables take on different values depending on whether
the snow is in the dry (d) or wet (w) regime.

Observations from the AC and ELA stations were used to determine $\alpha_{mx}$ and $\tau_x$. The optimal variables were found by
minimizing the weighted mean RMSE between the modelled and measured albedo by varying the values of $\alpha_{mx}$ and $\tau_x$. The
found best-fit values were $\alpha_{md}$=0.65, $\alpha_{mw}$=0.41, $\tau_{md}$=5 days, and $\tau_{mw}$=10 days.

Refreshment of the albedo to the maximum value only occurs if snowfall constitutes more than 95 % of the total precipitation.
It is possible to have a partial refreshment, as the albedo is only refreshed to the maximum allowed value if the amount of





snowfall on that day ($S_0$) is higher than 0.03 m w.eq. This value was chosen to provide the best fit with the AWS observations. The rate of refreshment $b$ is given by

$$b = min\left[1, \frac{S_f}{S_0}\right] \tag{3}$$

where $S_f$ is the amount of snowfall during the model time step in m w.eq. and $S_0$ is the critical amount of snowfall in m w.eq.
5  per model time step needed to completely refresh the albedo. Using this rate, the albedo is then refreshed using

$$\alpha_{snow}^{t+1} = \alpha_{snow}^t + b \cdot (\alpha_{max} - \alpha_{snow}^t) \tag{4}$$

where $\alpha_{max}$ is the maximum albedo for freshly fallen snow, set equal to 0.85 as this provides the best average fit with the observations.

In the case of small snow depths, the surface albedo will be affected by the albedo of the underlying ice. A smooth transition
between the snow and bare ice albedo is therefore implemented, and the final albedo is thus expressed as

$$\alpha^{t+1} = \alpha_{snow}^{t+1} + (\alpha_{ice} - \alpha_{snow}^{t+1}) \cdot exp(\frac{-d^{n+1}}{d_s}) \tag{5}$$

where $d$ is the snow depth, and $d_s$ is a characteristic scale for snow depth. Following Oerlemans and Knap (1998), the characteristic scale is set to 3.2 cm snow depth. If no snow is present, the albedo is set to the bare ice albedo.

In order to determine the bare ice albedo at each gridpoint, daily MODIS data over Iceland from 2001-2012 were used. Years
with volcanic eruptions were discarded, as the volcanic ash lowered the albedo values far below the average. The minimum autumn albedo value was then determined in each grid point and that value used to create a bare ice albedo map of the glaciers. The spectral properties of ice in the ablation zone are controlled by tephra layers in the ice, which are exposed as the glacier melts (Larsen et al., 1996). New falling tephra or dust will therefore only have a small effect on the spectral properties of the ice, as the ice surface is already covered in dark bands. In addition, the new particles are generally washed off from year to
20  year. Applying one background map for the entire period should therefore provide the same results as applying a map created for each year. In addition, it allows us to run the model for years where no MODIS observations are available or where the amount of observations over the ice cap are sparse due to e.g. clouds.

How much this bare ice MODIS albedo map improves the simulations will be estimated by comparing the results with those from a model simulation using a constant ice albedo in Section 4.8.

**3.2.2 Experimental design**

In this study, HIRHAM5 is run at a resolution of 0.05° on a rotated pole grid (equivalent to ~5.5 km). The model uses 31 irregularly spaced vertical atmospheric levels from the surface to 10 hPa with a model time step of 90 seconds in the dynamical scheme. The model is configured for a domain containing all of Greenland and Iceland. The model is forced at the lateral and





lower boundaries by the ECMWF ERA-Interim reanalysis dataset (Dee et al., 2011), which uses observations from satellites, weather balloons, and ground stations to create a comprehensive reanalysis of the atmosphere. The model is forced by temperature, wind, relative humidity and surface pressure at the lateral boundary, and sea surface temperature and sea ice fraction at the lower boundary at 6 hour intervals. The model is here run from 1980 to 2014.

The new snow/ice surface scheme discussed above is run offline in this study, meaning that the subsurface scheme is run separately from the atmospheric code. This is done by forcing the subsurface scheme every 6 hours by radiative and turbulent surface fluxes, as well as snow, rain, evaporation, and sublimation data from a HIRHAM5 experiment with a previous version of the albedo and refreezing schemes. While a full, high-resolution HIRHAM5 run is computationally very expensive, the offline model offers a fast and flexible option to test new model implementations and allows a quick and thorough spin-up of the subsurface. Running the model for a smaller domain which only contains Iceland further reduces the computational cost of the model. The reported values of albedo, upward longwave and shortwave radiation, and surface mass balance in the following are all from the offline run.

A disadvantage of this method is that it neglects feedbacks between the atmospheric circulation and the surface conditions like e.g. the albedo and temperature. However, since the surface temperature of Vatnajökull is typically near the melting point during the summer, both in reality and in the model, changes in the albedo should not have a large effect on the upward longwave radiation and the turbulent fluxes. Thus while the updated surface scheme is important for the mass balance components, the error due to the neglected feedbacks is likely small in the model calculations.

The offline model was initialized with values from a previous offline model run with a different albedo scheme and then a model spin-up was performed by integrating the model for 150 years repeating the forcing from 1980. The largest adjustments occurred during the first 75 years of the spin-up, after which the variation was much smaller than the interannual variability. At the end of the run, the solar radiation, surface mass balance, runoff, snow depth, and refreezing had all converged, as had the temperature, liquid and snow content in all 25 subsurface layers. The final state of the spin-up was then used as the initial condition for the 1980-2014 model simulation.

### 3.2.3 Elevation-based corrections

HIRHAM5 uses an elevation model over Iceland which has been interpolated onto the 5.5 km model grid. Since errors in the elevation of the glacier surface can introduce significant biases in temperature and pressure which are not caused by physical model errors (Box and Rinke, 2003), any elevation bias in the model has to be taken into account. The elevation bias was calculated as the difference between the model elevation, which was interpolated to the AWS sites using bilinear interpolation of the elevation at the four surrounding model grid points, and GPS observations at each site (Table 1).

The temperature was corrected for the elevation bias using a constant lapse rate of 6.5 °C km$^{-1}$, which resulted in temperature corrections on the order of 0.1-0.3 K. Correcting for the pressure was done using Eq. (1) and amounted to corrections on the order of 1 to 5 kPa. Thus although the HIRHAM5 elevation is consistently overestimated, it is not large enough to introduce significant biases in temperature and surface pressure.

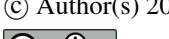



### 3.2.4 Model uncertainty

Due to nonlinearities in the HIRHAM5's model dynamics and physics, it has an implicit uncertainty due to internal model variability originating from nonlinear processes (e.g. Giorgi and Bi, 2000; de Elía et al., 2002). This variability is caused by numerical sensitivity, uncertainty in the boundary and initial conditions, and errors due to model parametrizations (e.g. Box and Rinke, 2003), like for example that of the albedo, the vertical gradients in the boundary layer, or cloud radiative effects. In addition, using a constant value of $z_0$ for both snow and bare ice could lead to large errors in the turbulent fluxes (e.g. Brock et al., 2000).

### 3.2.5 Validation method

AWS data from 2001-2014 for three Brúarjökull stations and two Tungnaárjökull stations are considered, as well as SMB point measurements from 1995-2014. All stations were operated during the summer months, but since 2006 the lowest Brúarjökull station has been operated year round. Comparisons are made between daily averages from the HIRHAM5 model and the in situ observations made by the AWSs. HIRHAM5 daily means are calculated from 6 hourly outputs, while the AWS daily means are calculated from observations at 10 minute intervals.

Comparisons between station values and model values are made by bilinear interpolation of the model output to the measurement position using the four closest model grid points and using only glacier-surface type grid cells.

In order to remove the effect of seasonally varying magnitudes of the components, the percent errors given in this study are calculated as the Root Mean Square Error (RMSE) divided by the observations.

## 4 Results and discussion

### 4.1 Meteorological variables

Before validating the surface energy balance components, a comparison of four near-surface variables with observations was made in order to assess how well they are simulated in the model. As the sensible and latent heat fluxes are computed using the surface pressure, $p_{sl}$, temperature, $T_{2m}$, relative humidity, $r_{2m}$, and wind speed, $w$, these model variables were evaluated at all five stations at the measurement height. How well these variables are simulated should indicate the model's ability to simulate the turbulent fluxes.

The comparison of the mean daily values during the summer months from 2001-2014 with corresponding observations from the five stations is shown in Table 2. The surface pressure, $p_{sl}$, which was not observed at the stations but estimated using Eq. (1), is generally forecast with a high degree of skill, with only a small error. At every station there is a high positive correlation between the HIRHAM5 simulated pressure and the pressure estimated using data from nearby meteorological stations (Eq. 1), with correlation coefficients higher than 0.9 both for the entire time series and for each year.

The model also captures the 2 m temperatures, $T_{2m}$, satisfactorily. The largest deviation from the observations is found at the $B_{AB}$ station, which underestimates the temperature with 0.8 K overall. The temperature is also underestimated at the four





remaining stations, but with less than 0.6 K. The model simulates the variation in temperature well; it for example captures the temperature dampening over a melting glacier surface. This is expressed in the high correlation values for all five stations.

The measured relative humidity, $r_{2m}$, at all five stations is generally high, with only 1-3 % of the data points at each station falling below 70 %, and the minimum daily value between 42 and 58 %. The model simulates a lower mean humidity than

the measured at all five stations, with 8-20 % of the points at each stations having values lower than 70 % and minimum daily values between 18 and 30 %. Since the exchange coefficient for moisture is a function of the atmospheric temperature profile, the underestimation of the relative humidity could be due to a too low temperature gradient. This is consistent with the underestimation found in the 2 m temperature. The correlation of between 0.68-0.7 indicates that the model simulates the humidity fluctuations satisfactorily.

Since the lowest wind speed level in HIRHAM5 is at 10 m and the AWS wind speeds are measured at between 2-4 m, depending on the year, the HIRHAM5 wind speed is extrapolated to the measurement height using a logarithmic profile with a roughness length of 1 mm. At all five locations, HIRHAM5 simulates winds that are too weak on average. This could be due to the uncertainty arising from the interpolation of the model winds from second-lowest level (30 m) to the lowest level (10 m) under stable conditions, as the wind speed can change significantly over the 20 m interval.

## 4.2 Longwave radiation

As shown above, HIRHAM5 underestimates the temperature at all five stations, with the largest underestimation at the $B_{AB}$ station. Therefore, it is to be expected that the incoming longwave radiation is underestimated at all five stations, with the largest difference occurring at the $B_{AB}$ station. As shown in the scatter plots in Figure 2a, this is indeed the case. The average percentage difference is approximately 8 % for all five locations (see Table 3), so on average the results are within the 10 %

uncertainty of the AWS observations. However, as can be seen in the figures, many of the simulated days have larger errors - between 25-30 % of the simulated points have percent difference larger than 10 %.

The incoming LW radiation is mainly emitted from clouds and atmospheric greenhouse gases, and therefore a source of the underestimation could be either that the model underrates cloud formation and/or simulates clouds that are too optically thin in the LW region of the spectrum. An underestimation of the temperature in the atmosphere could also be causing the

underestimation.

Figure 2b shows the comparison of the modelled and measured outgoing LW radiation. There is a small overestimation at the $T_{AC}$ station, and a small underestimation of the other four stations, but in general the model reproduces the daily values well. The average percentage deviation between the modelled and measured values is only around 3 %, and only between 0.5-2 % of the HIRHAM5 data points have deviations larger than 10 %.

Due to an underestimation of the incoming LW radiation, and only small negative or positive biases in the outgoing LW, the total LW radiation has a mean negative bias at all AWS locations. The average model error for all five stations is -7.9 W m$^{-2}$.



### 4.3 Shortwave radiation and albedo

Figure 3 and Table 3 show the comparisons of the modelled and measured components of the shortwave (SW) radiation as well as the surface albedo. On average, the incoming SW radiation is underestimated at all five stations. This underestimation is also present in the means at all five stations for most years, except in 2002, 2004, 2005, and 2014 at the $B_{AB}$ station. This suggests that there are errors in either the modelling of the clouds, e.g. due to an overestimation of the cloud fraction, the amount of cloud formation, or the optical thickness of the clouds in the shortwave region, and/or because of errors in the clear-sky fluxes.

The albedo comparison is shown in Figure 3b. The modelled albedo at the two AB stations has the largest deviation from the observations; this is partly due to the modelled snow cover, which either does not completely disappear or disappears later in the year than the AWS data show. At the $B_{AB}$ station, the ice layer is generally exposed in the model, although the ice surface is always exposed later than in reality. One exception is in 2001, where the albedo never drops to the ice value in HIRHAM5, but the observations show albedo values down to 0.03. This one year therefore highly contributes to the total overestimation of the albedo. Comparisons with the mass balance measurements (discussed below in Section 4.6.1) show that the winter balance is overestimated during approximately half of the study period, which will contribute to the too slow albedo drop in the model.

At the $T_{AB}$ station, the modelled winter snow cover is also the cause of some of the discrepancy. Here, the ice surface is not exposed in the model during any of the modelling years, and the albedo never drops much below 0.4 (the minimum snow albedo), even though the AWS data shows that the ice surface was exposed during all but two years. However, during these two years where the ice surface was not exposed, the simulated albedo fits well with observations. This station is only ∼100 m below the average ELA, and thus some of the same modelling difficulties which affect the ELA station (discussed below) may be found on this station, especially during years where the snow line is closer to the station. In addition, comparisons with mass balance measurements (Section 4.6.1) show that the winter balance is always overestimated at this station, and an overestimation of the snow layer at the beginning of summer, combined with an underestimation in the radiation and turbulent fluxes, is the likely reason for the overestimation of the snow layer at the end of summer.

Another issue which affects both stations is that the MODIS albedo at these points is not as low as the measured albedo. The MODIS ice albedo at these stations is 0.10 ($B_{AB}$) and 0.16 ($T_{AB}$), whereas the observations show the albedo can drop as low as 0.01 at both stations. The albedo drops below the MODIS value every year at the $B_{AB}$, and during 2001-2005 and 2011 at the $T_{AB}$ stations. This is presumably due to the heterogeneity of the albedo in the ablation zone, which means that low in situ albedo value at a point cannot be captured at the current HIRHAM5 resolution.

At the ELA station, the mean albedo value is underestimated (Table 3). This is due to the difficulty in modelling the albedo near the equilibrium line. In this area, the albedo is highly variable both temporally and spatially. There is e.g. a large difference in albedo depending on whether the previous year's summer surface was exposed or not. In general, the model overestimates the albedo during years where the summer surface was exposed, and underestimates the albedo during years where it was not. In addition, the winter mass balance at this station is always underestimated (Section 4.6.1), meaning the thickness of




snow layer in spring is underestimated and the effect of the underlying ice layer will therefore be overestimated, leading to the underestimation in albedo.

The difference between the model and the observations is smallest at the two AC stations. The $B_{AC}$ station generally provides the best fit with the observations, while the model tends to underestimate the albedo at the $T_{AC}$ station. An exception to this is in 2010 and 2011, where the albedo was overestimated by the model at both stations due to ash deposition from the Eyjafjallajökull and Grímsvötn eruptions (e.g. Gudmundsson et al., 2012).

A general reason for the model underestimating the albedo is that it does not take the albedo changes due to dust storms or volcanic dust deposition into account. One example of this is the very low albedo values seen in Figure 3b at the $T_{AC}$ station (blue) which are due to tephra deposited on the glacier during the 2010 eruption of Eyjafjallajökull (e.g. Gudmundsson et al., 2012; Gascoin et al., 2017). Even though dust events do not cause as large changes in albedo as a volcanic eruption, they can still significantly lower the albedo (e.g. Painter et al., 2007; Dragosics et al., 2016) . As previously mentioned, the albedo in HIRHAM5 often reaches its yearly minimum value later in the summer than the observed, and dust events, which cause the albedo to drop faster or earlier in the year than would otherwise be the case, could explain this discrepancy. A study by Dragosics et al. (2016) for example investigated 10 dust events which occurred at the $B_{ELA}$ station in 2012, and found a lowering in the albedo during all events and that the dust storms had a significant effect on the resulting energy balance.

The error in the outgoing shortwave radiation is, of course, caused by errors in the albedo and the incoming SW. At the $B_{AB}$ station, the incoming radiation is slightly underestimated but the albedo is overestimated, hence the outgoing SW is overestimated. The values at the four other stations are all underestimated, due to larger underestimations of the incoming SW radiation and lower albedo errors.

Due to the underestimation of both the incoming and outgoing SW radiation at most stations, the two components partially offset each other. The net SW at the three highest stations is generally underestimated by between $\sim$-6 and -12 W m$^{-2}$, while the values at the two AC stations are underestimated by between -22 and -28 W m$^{-2}$. The average model error at all five stations is therefore negative at -15.5 W m$^{-2}$.

### 4.4 Turbulent fluxes

Based on the comparison of the measured and modelled meteorological variables, one would expect HIRHAM5 to forecast the turbulent fluxes with a significant model underestimation. As can be seen in the comparisons statistics in Table 3 and the scatterplots in Figure 4, this appears to be the case. The two AC stations have the largest differences and also the lowest correlation (0.45 and 0.49) between the AWS model and the HIRHAM simulation. The other three stations also have significantly lower values in the HIRHAM5 model than in the AWS model, but with higher correlation coefficients (0.69-0.73).

It is important to bear in mind that this comparison is a model-model comparison, so while the eddy flux model may give a good estimate of the turbulent fluxes, model errors still affect the results e.g. due to the use of a constant roughness length.



## 4.5 Total energy balance

After the simulated components of the energy balance were evaluated against AWS observations, the total energy was estimated (see Table 3). Overall, the energy balance is underestimated, owing to all elements of the energy balance generally being underestimated. This is in large part due to the underestimation of the modelled incoming radiation. We attribute this to an error in the modelling of the clouds, but since both the incoming SW and LW radiation are underestimated, the cloud cover cannot be the only source of the error. An error in the cloud cover would have to be combined with an error in the interaction between clouds and radiation, for example in the form of an error in the optical thickness of the clouds, or errors in the clear sky fluxes, to explain these results. The underestimation of the incoming LW radiation could also be due to errors in the vertical atmospheric temperature gradient.

Whereas the simulated outgoing LW radiation generally only has a small deviation from the measured, and the deviation in the LW radiation is therefore mostly due to the incoming radiation, the errors in the simulated albedo means that both the in- and outgoing SW radiation greatly contribute to the deviation in the total SW radiation. Some of the error can be attributed to ash and dust deposition during volcanic eruptions and dust storms, which the model does not take into account and we therefore cannot expect these events to be reflected in the model results. However, as mentioned above, another issue with the simulated albedo stems from a snow layer that disappears too slowly compared to AWS results at stations in the ablation zone, which means that the simulated albedo drops too slowly compared to the measured albedo. The underestimation of the net radiation and the turbulent fluxes also contributes to the overestimation of the modelled snow layer, as they all contribute to the underestimation of energy available for melt.

In order to estimate how much the different components contribute to the energy difference on a year-to-year basis, the mean difference of the energy components during each summer (Apr-Oct) was plotted for each station (Fig. 5).

At the $B_{AC}$ station, the contribution of the long- and shortwave radiation and turbulent fluxes to the energy difference is consistent for the entire period, with the error of each component being almost equal, varying between -25 and 0 W m$^{-2}$. At the $T_{AC}$ station, the error due to the three components is also of the same order of magnitude, except in 2010 and 2011 where there error in the SW radiation is much larger than that in the other components. This is due to a large drop in the albedo as a result of the Eyjafjallajökull (2010) and Grímsvötn (2011) eruptions. The mean difference in the SW radiation for non-eruption years is -3 W/m$^2$, whereas the radiation difference in 2010 is -106 W m$^{-2}$. Assuming the deviation from the mean is only due to the volcanic eruption, the contribution to the energy is -103 W m$^{-2}$ over a 128 day measurement period. If it is further assumed that the surface was always at melting point, the increase in melt due to the 2010 Eyjafjallajökull eruption would be ∼3.1 m w.eq. at this station.

At the ELA site, the contribution from the modelled turbulent fluxes to the energy balance deviation generally varies between ±10 W m$^{-2}$, except in 2013 where the bias is around -25 W m$^{-2}$. The difference in modelled and measured longwave radiation is consistently at about -10 W m$^{-2}$. The deviation in the shortwave radiation is more variable, as expected from the results of the albedo comparison; depending on whether the previous years summer surface was exposed or not, the albedo was generally





either over- or underestimated. For example, at $B_{ELA}$, the summer surface was reached in for e.g. 2007 and 2012, resulting in an overestimation of the albedo. In e.g. 2002 and 2009, however, the albedo was high the entire summer as the previous summer surface was not exposed, resulting in an underestimation of the predicted albedo.

At the $T_{AB}$ station, both the LW radiation and the turbulent fluxes agree well with observations for the entire period. The
5 SW radiation, however, is always underestimated, especially in 2001-2003 and 2011. These years, the measured albedo at the station goes below 0.1, while the HIRHAM5 albedo stays around 0.4. As previously discussed, this albedo bias occurs because of an overestimation of the snow cover at the station due to an overestimation of the winter accumulation and possibly also the proximity to the equilibrium line, and it is the main reason for the large underestimation of the SW radiation. An underestimation of the incoming SW radiation, which we attribute to an error in cloud cover amount of clear-sky fluxes, also
contributes to this error.

At the $B_{AB}$ station, the longwave radiation bias is relatively constant with values close to 0 W/m² for much of the measurement period. The absolute deviation due to the turbulent fluxes is less than 10 W m⁻² for most of the period, although with slightly larger deviations from 2007-2010. The SW radiation is always underestimated at this station, mostly due to the previously discussed overestimation of the albedo.

## 4.6 Surface mass balance

### 4.6.1 At AWS sites

Scatter plots of measured and HIRHAM5 simulated SMB are shown in Figure 6 and the average deviations are shown in Table 4.

The winter mass balance comparison offers an evaluation of the winter precipitation in HIRHAM5. The simulated mass
balance at the $B_{ELA}$, $B_{AC}$, and $T_{AC}$ stations is underestimated during all years but one, while the simulated value at the $T_{AB}$ station is overestimated over the whole period. The modelled mass balance at the $B_{AB}$ station has an almost equal amount of years which are over- and underestimated. Apparently the model either carries too much precipitation when the clouds reach the glacier, resulting in too much precipitation at the ice sheet margin, or more melting occurs at the ablation area stations during the winter months than the model estimates.

The summer SMB results are in good agreement with the results of the energy balance calculations. The summer SMB is generally overestimated, although it is underestimated occasionally at all stations except $T_{AB}$. The ELA station has the largest amount of underestimated points, which is consistent with the findings from the energy balance calculations. Besides the errors introduced due to the underestimation of the energy balance, possible over- or underestimations of the modelled summer accumulation contribute to these errors as well.

Due to the difference in the summer and winter balance, the net balance at the $B_{AC}$, $T_{AC}$, and $B_{ELA}$ stations is generally underestimated in HIRHAM5, while the balance at the two AB stations is generally overestimated. This is due to a general overestimation of the winter balance in the ablation area, either due to an underestimation of the winter melt or an overestimation of precipitation, as discussed above.



### 4.6.2   At all measurement sites

The mass balance was not just measured at the AWS sites, but at between 25-120 sites on the ice cap, depending on the year (Fig. 1). In order to estimate how well the model represents the SMB at non-AWS sites, the data from all the sites between 1995-2014 were compared with the HIRHAM5 simulation (Fig. 7; Table 4).

The winter balance at all measured points is slightly overestimated by HIRHAM5 on average. However, this is mostly due to a large difference between measured and simulated SMB at the ice covered, high elevated, central volcano Öræfajökull (the white dots in Fig. 7). Only one site has been measured on this glacier for a few years only (Guðmundsson, 2000), in a spot that always receives a large amount of accumulation, but since HIRHAM5 consistently overestimates the accumulation by 100-200 %, this one point has a large effect on the mean error. This is a well known issue with hydrostatic models like HIRHAM5, as

they characteristically overestimate the precipitation on the upslope and peaks in complex terrain. The reason for this is that the precipitation is calculated as a diagnostic variable, i.e. it is not governed by an equation that is a derivative of time, meaning that when the required conditions for precipitation are met in the local atmosphere, the precipitation appears instantaneously on the surface. Thus the scheme does not allow horizontal advection of snow and rain by atmospheric winds, which is a particularly significant process in complex terrain, as it can force the precipitation downslope (e.g. Forbes et al., 2011). Without this effect,

precipitation is generally overestimated at high peaks like Öræfajökull. If this point is removed from the comparison, the total difference is about one-third that of the AWS sites only (-0.09 m w.eq.). This is due to the sites closer to the edge of the ice cap, as winter balance at the measurement points at the outer parts of the icecap generally is overestimated in the model, while the balance at the sites in the middle is underestimated.

On average, the summer ablation is underestimated, which is consistent with the findings from the AWS stations that there

is an average underestimation of the energy available for melt. The mean error and RMSE is only slightly larger than at the AWS sites.

The mean net balance is overestimated by approximately the same amount as the summer balance, partly due to the low mean deviation in the winter SMB. Due to the large deviation at Öræfajökull in the winter SMB, the Öræfajökull points clearly have the largest bias. If these points are excluded, a rmse closer to that for the AWS locations is found (1.1 m).

### 4.7   Reconstructing the SMB of Vatnajökull

Having assessed how well the model simulates the energy and mass balance components at the measurement sites, the model was then used to estimate the mean specific mass balance of Vatnajökull back to 1981. The specific winter, summer, and net mass balances of Vatnajökull were calculated for the entire simulation period, and the results were compared with an estimate of the specific balance from 1995-2014, created by manual interpolation of the mass balance measurements (e.g. Pálsson et al.,

2015), see Fig. 8. The model prediction of the mean specific summer mass balance generally fits well with the interpolated observations, with an overall difference of only 0.06 m w.eq. The largest deviations are in 1995, where there is too much ablation in the simulation, and in 1997, 2005, and 2010-2012, where there is too little ablation in the simulation, most likely




due to ash depositions on the glacier following the 1996 Gjálp eruption, the 2004 and 2011 Grímsvötn eruptions or the 2010 Eyjafjallajökull eruption, which are not taken into account in the model.

Excluding the years where the albedo was affected by volcanic eruptions, the average difference becomes smaller but the model also predicts slightly too much ablation, as the difference becomes -0.02 m w.eq.

The specific winter mass balance is larger in HIRHAM5 for the entire measurement period with an average of 0.54 m w.eq. Due to this difference, and only the small negative mean difference in summer mass balance, the annual mass balance of Vatnajökull is overestimated every year with an average difference of 0.50 m w.eq.

However, this is mostly due to the large overestimation of the winter accumulation on Öræfajökull; comparison with the mass balance measurements showed that the model overestimated the winter accumulation by 100-200 % compared with the

obervations. In an attempt to estimate how much this error affects the results, a simple correction was added to the Öræfajökull points by reducing the simulated winter SMB by 50 %. The correction was added to four model grid points around Öræfajökull, due to the high (>10 m/yr) annual specific mass balance in these points (see Fig. 10a). The resulting modelled winter and annual specific balance are shown in Fig. 9. The winter balance is still overestimated, but the difference between modelled and interpolated values has been reduced to only 0.1 m w.eq. In addition, the average difference between the HIRHAM5 and

interpolated annual SMB drops to only 0.08 m w.eq.

Spatial maps of the (uncorrected) average winter, summer, and net SMB from the 1980-81 glaciological year until 2013-14 are shown in Figure 10. The average deviation between observation and model over the observation period at each measurement location is also shown, in order to give an indication to the average error of the model at different parts of the ice cap.

### 4.8   Comparison with constant ice albedo simulation

In order to quantify the changes in the model performance resulting from the new albedo scheme used in this study, which utilizes an albedo map based on MODIS data (Gascoin et al., 2017), the results are compared to those of a run without MODIS albedo, which uses an ice albedo of 0.3. The average difference in albedo and mass balance over the period 2001-2014 in each grid point are shown in Fig. 11, as well as the position of the AWS stations.

There is little to no difference between the two runs in the accumulation zone, due to the year-round snow cover. In the

ablation zone, however, using the MODIS ice albedo map has a large effect on the simulated albedo. The largest difference appears to be on the southern outlet glacier Skeiðarárjökull, which is unfortunately a glacier where no mass balance or AWS measurements have been conducted. The $B_{AB}$ and $B_{ELA}$ stations are in areas that are affected by the ice albedo, either because ice is exposed ($B_{AB}$) or because the underlying surface contributes to the albedo ($B_{ELA}$). The $T_{AB}$ station is in the ablation area, but for example due to an overestimation of the winter accumulation (Section 4.6.1) the ice surface is never exposed in

the model. The albedo estimate at this station was therefore not improved by using the MODIS albedo.





## 5 Conclusions

The comparison of a HIRHAM5 simulation with data from five AWSs on Vatnajökull ice cap allowed us to draw valid conclusions about the model performance. By comparing observations from April-October with model output, it was found that the model simulates the surface energy balance components and surface mass balance well, albeit with general underestimations.
Even though the energy balance was generally underestimated, the model simulated the near-surface temperature well. The reason for this is that the comparisons only uses observations from the summer months, where the glacier surface is generally at the melting point, and thus the energy is used for melting and not for raising the temperature of the surface.

The modelled incoming radiation is underestimated on average in both the shortwave and longwave spectrum, which we suggest is due to biases in the modelling of the cloud cover combined with errors in the optical thickness in the short- or longwave spectrum, or errors in the clear-sky fluxes.

Whereas the modelled outgoing LW radiation component is within the uncertainty of the LW observations at the five stations, suggesting that the model simulates the surface temperatures well, there was a larger differences between the modelled and measured outgoing SW radiation. This is partly due to the underestimation of the incoming SW radiation and partly due to inaccuracies in the simulated albedo. The albedo, which was simulated using an iterative, temperature based albedo scheme (Nielsen-Englyst, 2015) with a bare ice albedo determined from MODIS data (Gascoin et al., 2017), was shown to provide a better fit with AWS albedo observations than when using a constant ice albedo. However, the simulated albedo was generally overestimated during the summer and did not reach the lowest yearly value as early in the year as the measured albedo, particularly in the ablation zone. This was attributed to an overestimation of the snow cover in the ablation zone, an overestimation in the MODIS ice albedo compared with AWS observations, and that the model does not account for the effect of volcanic dust deposition during eruptions and dust events on the albedo. A possible way to include dust storms or eruptions into the model could be including a stochastic ashes or dust generator, which distributes dust onto the glacier. Including simulations of dust depositions and concentrations from a dust mobilization model could also be an option, as e.g. Dragosics et al. (2016) used the model FLEXDUST to simulate dust events on Vatnajökull in 2012, and found that the modelled dust events correspond well with albedo drops at two AWSs on Brúarjökull.

Due to the general underestimation of the energy balance components, the ablation during the summer months is underestimated on average. Comparison with mass balance measurements from the AWS sites and from sites scattered across Vatnajökull shows an overall overestimation of the summer balance by about 0.5 m w.eq. The overestimation is largest in the ablation zone. The winter balance is on average underestimated at the survey sites, albeit with the highest measuring site (on Öræfajökull) having a large overestimation of the winter balance.

The mean specific summer, winter and net mass balances were reconstructed for all of Vatnajökull from 1981-2014 and compared to estimates of the mass balance from 1995-2014. The summer balance is overestimated with 0.06 m w.eq. on average, i.e. there is generally too little ablation in the summer, with too much ablation in 1995 and too little ablation in years with, or following, volcanic eruptions. The winter balance is overestimated by 0.5 m w.eq., mostly due to a large overestimation at the high elevation glacier Öræfajökull. This overestimation of accumulation at high elevation is characteristic for hydrostatic



RCMs (Forbes et al., 2011). If the overestimation at these points is corrected, we estimate that the simulated winter balance would fit well with the observations, as the overestimation of the balance would drop to around 0.1 m w.eq..

There is a shift in the summer and annual mass balance calculated by the model and the in situ MB measurements around 1996, with a generally more negative mass balance after 1996 than before. This is consistent with the increase of the annual mean temperature of Iceland in the mid-1990s, which resulted in a mean annual temperature $\sim 1\,°C$ higher in the decade after than the decade prior to 1995. This is likely linked with atmospheric and ocean circulation changes around Iceland, as there was a rapid increase in ocean temperatures off the southern coast in 1996 (Björnsson et al., 2013). That the model catches the changes in the specific mass balance well over the mass balance measurement period, and also captures the shift in mass balance in the mid-90s, gives us confidence that the model estimates the specific mass balance of Vatnajökull well over the entire simulated period from 1980-2014. The model is therefore a useful tool to expand the time series of the specific SMB beyond the measurement years. However, as ERA-Interim reanalysis data only goes back to 1979, the model would need to be forced at the lateral boundary by another dataset in order to estimate the mass balance before 1980, like for example ERA-20C (Poli et al., 2016). However using other reanalysis data probably leads to different errors; this needs further investigation. The model could also be a useful tool to estimate the future evolution of the SMB of the ice cap, but this would also require a different forcing at the lateral boundary like general circulation model output. This would most likely introduce larger biases than the ones found using ERA-Interim, and the magnitude of these biases would need to be estimated and corrected before using the model for future projections.

## 6    Data availability

HIRHAM5 output is freely accessible from http://prudence.dmi.dk/data/temp/RUM/HIRHAM/GL2/, as is MODIS data from https://modis.gsfc.nasa.gov/data/. Measurements from automatic weather stations and from in situ mass balance surveys are partially owned by the National Power Company of Iceland and are therefore not publicly available at this time.

*Competing interests.*   The authors declare that they have no conflict of interest

*Acknowledgements.*   This work is supported by project SAMAR, funded by the Icelandic Research Fund (RANNIS, Grant no. 140920-051), as well as the National Power Company of Iceland (Landsvirkjun). Measurements from automatic weather stations and in situ mass balance surveys are from joint projects of the National Power Company and the Glaciology group at the Institute of Earth Sciences, University of Iceland.





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



## Tables and figures

**Table 1.** Average measured elevation and average bias of the interpolated HIRHAM5 elevation at each station for 2001-2014.

| Station | Average elevation [m] | Average model elevation bias [m] |
|---------|------------------------|-----------------------------------|
| $B_{AB}$ | 839 | 22 |
| $T_{AB}$ | 1089 | 47 |
| $B_{ELA}$ | 1205 | 31 |
| $B_{AC}$ | 1526 | 17 |
| $T_{AC}$ | 1457 | 13 |





**Table 2.** Comparison of HIRHAM5 and AWS during the summer months (Apr-Oct) for the period from 2001-2014. The HIRHAM5 bias (HIRHAM5-AWS), the root-mean-sqaure error (rmse), the percentage error, and the correlation (r) are shown.

| Parameter | Station | HIRHAM5 value | HIRHAM5 bias | rmse | % error | r |
|---|---|---|---|---|---|---|
| $p_{sl}$ [hPa] | $B_{AB}$ | 911.7 | -0.2 | 2.8 | 0.3 | 0.96 |
| | $T_{AB}$ | 883.8 | -0.4 | 3.0 | 0.3 | 0.95 |
| | $B_{ELA}$ | 871.5 | -0.6 | 2.9 | 0.3 | 0.95 |
| | $B_{AC}$ | 837.1 | 0.1 | 2.2 | 0.3 | 0.97 |
| | $T_{AC}$ | 844.2 | -0.9 | 2.7 | 0.3 | 0.96 |
| $T_{2m}$ [K] | $B_{AB}$ | 273.3 | -0.8 | 1.5 | 0.6 | 0.94 |
| | $T_{AB}$ | 273.5 | -0.6 | 1.3 | 0.5 | 0.89 |
| | $B_{ELA}$ | 272.9 | -0.1 | 1.1 | 0.4 | 0.91 |
| | $B_{AC}$ | 271.5 | -0.1 | 1.4 | 0.5 | 0.90 |
| | $T_{AC}$ | 272.1 | 0.0 | 1.2 | 0.5 | 0.91 |
| $r_{2m}$ | $B_{AB}$ | 81.7 | -6.2 | 12.2 | 13.9 | 0.68 |
| | $T_{AB}$ | 83.5 | -6.1 | 11.5 | 12.9 | 0.76 |
| | $B_{ELA}$ | 88.0 | -3.8 | 9.8 | 10.7 | 0.73 |
| | $B_{AC}$ | 90.4 | -3.5 | 9.6 | 10.2 | 0.68 |
| | $T_{AC}$ | 87.4 | -2.6 | 9.7 | 10.7 | 0.72 |
| $u$ [m s$^{-1}$] | $B_{AB}$ | 3.9 | -1.2 | 2.0 | 39.0 | 0.80 |
| | $T_{AB}$ | 5.0 | -0.3 | 1.8 | 33.0 | 0.87 |
| | $B_{ELA}$ | 4.3 | -0.1 | 1.8 | 41.1 | 0.82 |
| | $B_{AC}$ | 5.2 | -0.7 | 1.8 | 30.8 | 0.86 |
| | $T_{AC}$ | 5.1 | -0.1 | 2.0 | 38.9 | 0.82 |



**Table 3.** Comparison of incoming and outgoing long- and shortwave radiation, albedo ($\alpha$), turbulent fluxes ($H_{s+l}$), and total energy ($E$) from HIRHAM5 simulations and AWS measurements during summer months (Apr-Oct) from 2001-2014. The HIRHAM5 bias (HIRHAM5-AWS), the root-mean-square error (rmse), the percentage error, and the correlation (r) are shown.

| Parameter | Station | HIRHAM5 value | HIRHAM5 bias | rmse | % error | r |
|---|---|---|---|---|---|---|
| LW↓ [W m$^{-2}$] | $B_{AB}$ | 273.8 | -16.9 | 26.3 | 9.1 | 0.79 |
| | $T_{AB}$ | 280.3 | -7.0 | 20.9 | 7.3 | 0.80 |
| | $B_{ELA}$ | 274.9 | -9.0 | 21.7 | 7.7 | 0.79 |
| | $B_{AC}$ | 271.8 | -8.5 | 24.4 | 8.7 | 0.79 |
| | $T_{AC}$ | 270.3 | -3.8 | 20.4 | 7.4 | 0.83 |
| LW↑ [W m$^{-2}$] | $B_{AB}$ | 307.3 | -1.9 | 7.3 | 2.4 | 0.87 |
| | $T_{AB}$ | 309.5 | -2.5 | 7.4 | 2.4 | 0.78 |
| | $B_{ELA}$ | 307.0 | -3.3 | 10.5 | 3.4 | 0.70 |
| | $B_{AC}$ | 298.4 | -1.5 | 12.9 | 4.3 | 0.76 |
| | $T_{AC}$ | 303.9 | 2.6 | 11.6 | 3.9 | 0.68 |
| SW↓ [W m$^{-2}$] | $B_{AB}$ | 185.2 | -4.0 | 55.5 | 29.3 | 0.81 |
| | $T_{AB}$ | 185.7 | -35.2 | 72.2 | 32.7 | 0.79 |
| | $B_{ELA}$ | 193.1 | -36.2 | 64.6 | 28.1 | 0.83 |
| | $B_{AC}$ | 193.0 | -43.7 | 69.9 | 29.5 | 0.82 |
| | $T_{AC}$ | 206.0 | -41.9 | 72.5 | 29.2 | 0.79 |
| SW↑ [W m$^{-2}$] | $B_{AB}$ | 104.7 | 18.1 | 61.0 | 70.4 | 0.64 |
| | $T_{AB}$ | 105.6 | -6.9 | 54.7 | 48.7 | 0.73 |
| | $B_{ELA}$ | 116.2 | -29.9 | 59.2 | 40.5 | 0.75 |
| | $B_{AC}$ | 141.9 | -31.3 | 56.4 | 32.6 | 0.79 |
| | $T_{AC}$ | 140.2 | -33.4 | 65.6 | 37.8 | 0.68 |
| $\alpha$ [%] | $B_{AB}$ | 47.3 | 12.7 | 23.6 | 68.2 | 0.75 |
| | $T_{AB}$ | 54.5 | 9.96 | 21.0 | 47.2 | 0.68 |
| | $B_{ELA}$ | 57.8 | -2.9 | 18.4 | 30.2 | 0.57 |
| | $B_{AC}$ | 73.0 | 0.8 | 10.5 | 14.5 | 0.62 |
| | $T_{AC}$ | 67.9 | -2.2 | 16.1 | 22.9 | 0.47 |
| $H_{s+l}$ [W m$^{-2}$] | $B_{AB}$ | 29.7 | -5.0 | 28.6 | 116 | 0.71 |
| | $T_{AB}$ | 32.4 | -3.8 | 25.2 | 69.6 | 0.79 |
| | $B_{ELA}$ | 22.5 | -2.0 | 26.2 | 107 | 0.71 |
| | $B_{AC}$ | 8.4 | -12.3 | 28.2 | 136 | 0.31 |
| | $T_{AC}$ | 14.5 | -6.3 | 23.0 | 110 | 0.49 |
| E [W m$^{-2}$] | $B_{AB}$ | 87.2 | -44.4 | 82.8 | 62.9 | 0.67 |
| | $T_{AB}$ | 83.4 | -36.7 | 98.0 | 72.3 | 0.58 |
| | $B_{ELA}$ | 71.0 | -13.4 | 49.6 | 58.8 | 0.68 |
| | $B_{AC}$ | 36.2 | -28.6 | 50.3 | 77.5 | 0.53 |
| | $T_{AC}$ | 46.5 | -21.2 | 78.6 | 89.7 | 0.43 |





**Table 4.** Comparison of HIRHAM5 and mass balance measurements, both at AWS sites and for all measuring sites on Vatnajökull.

|  | Season | HIRHAM5 value | HIRHAM5 bias | rmse | % error |
|---|---|---|---|---|---|
| AWS locations | Winter | 1.11 | -0.26 | 0.71 | 51.6 |
|  | Summer | -1.86 | 0.48 | 0.81 | -34.6 |
|  | Total | -0.75 | 0.23 | 1.15 | -118 |
| All locations | Winter | 1.50 | 0.04 | 1.21 | 82.9 |
|  | Summer | -1.76 | 0.52 | 0.94 | -41.1 |
|  | Total | -0.27 | 0.56 | 1.56 | -186 |



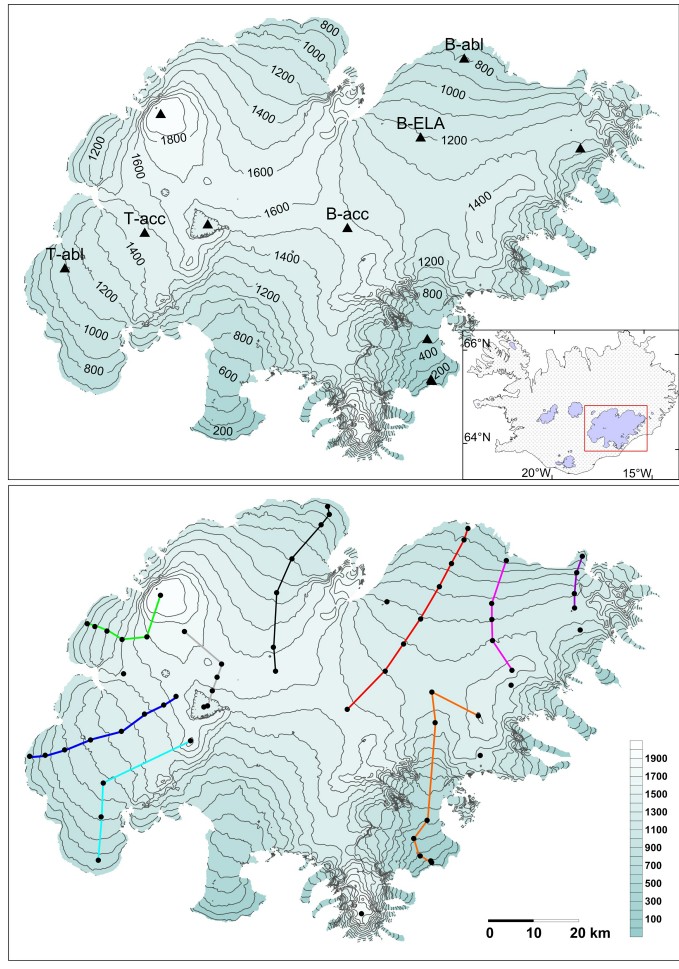

**Figure 1.** *a) the AWS sites and b) mass balance sites from 1995-2014. Not all mass balance sites were measured every year.*





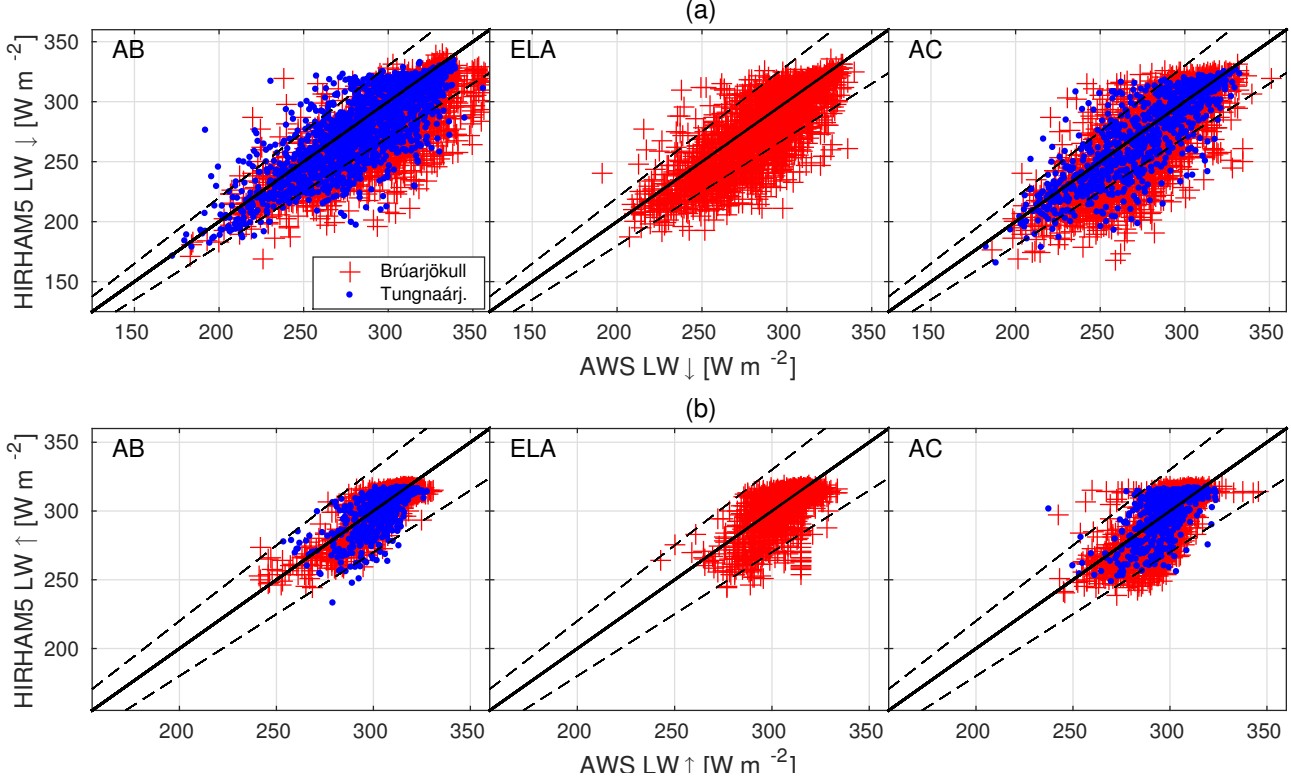

**Figure 2.** Scatter plots of the measured longwave radiation components, LW↓ and LW↑, by stations on Brúarjökull (red) and Tungnaárjökull (blue) versus the LW radiation components simulated by HIRHAM5 at the same locations. The dashed line corresponds to ±10%, *i.e.* the manufacturer reported uncertainty of the AWS measurements.





**Figure 3.** Scatter plots of the measured shortwave radiation components, a) SW↓, b) albedo, and c) SW↑, by stations on Brúarjökull (red) and Tungnaárjökull (blue) versus the shortwave radiation components simulated by HIRHAM5 at the same locations. The dashed line corresponds to the uncertainty of the measured AWS components.





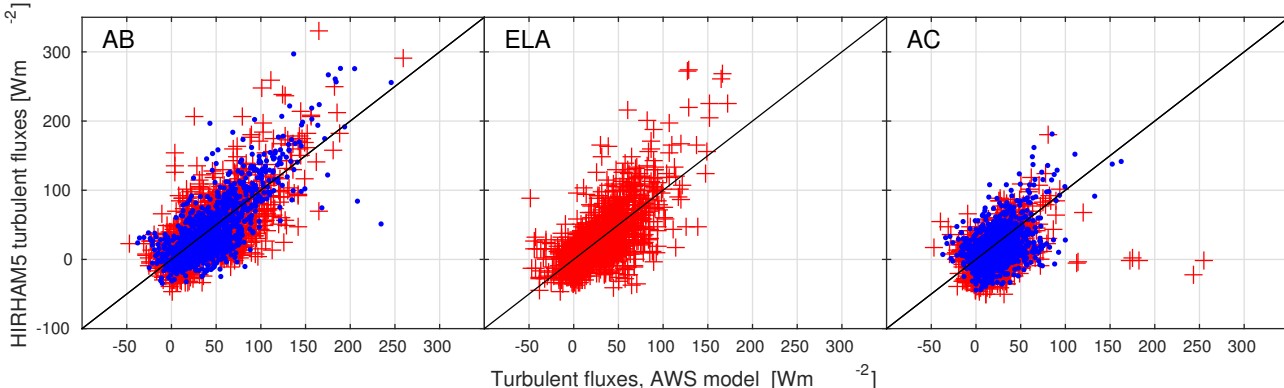

**Figure 4.** The total turbulent fluxes calculated using the a one-level flux model versus the HIRHAM5 simulated values. Red marks are the data from the Brúarjökull stations, and blue marks are for the Tungnaárjökull stations.





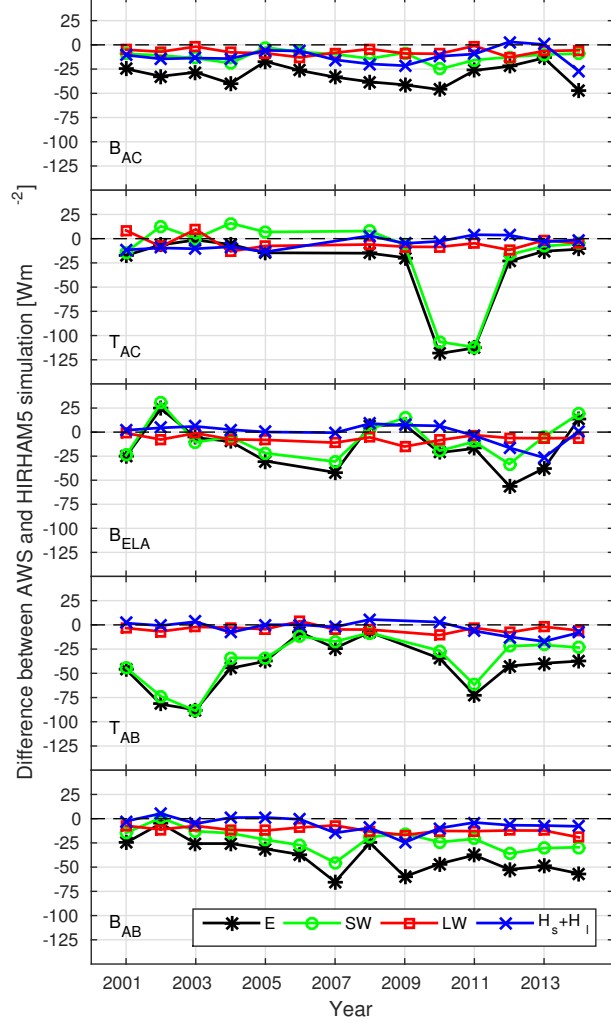

**Figure 5.** The average summer (Apr-Oct) bias of each energy balance component for the measurement period at each AWS site. The large deviation in the SW radiation at the Tunaárjökull sites in 2010-2011 is due to deposition of ash on the glacier during the 2010 Eyjafjallajökull and 2011 Grímsvötn eruptions.



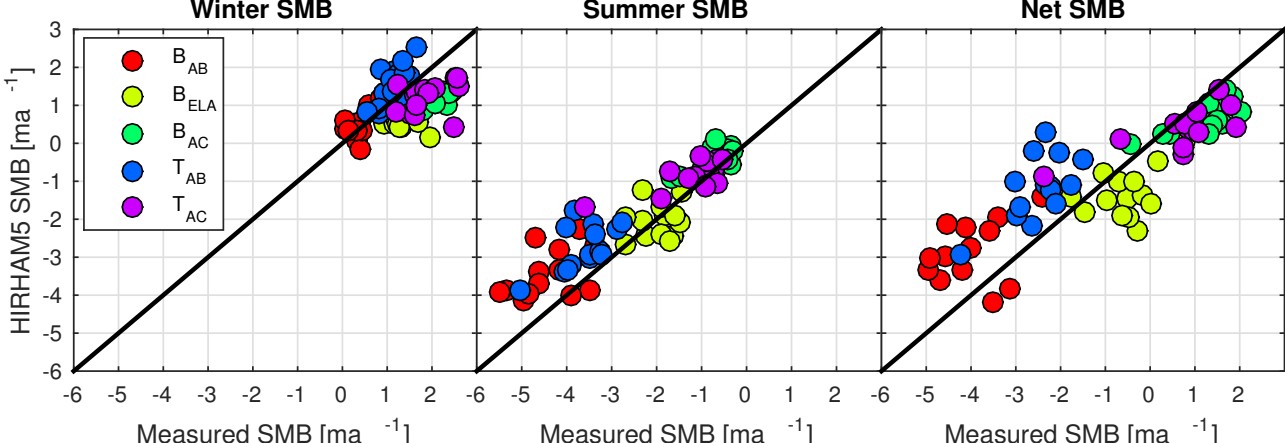

**Figure 6.** *The winter, summer, and net mass balance from 1995-2014 according to mass balance measurements at the five AWS sites and the HIRHAM5 simulation.*





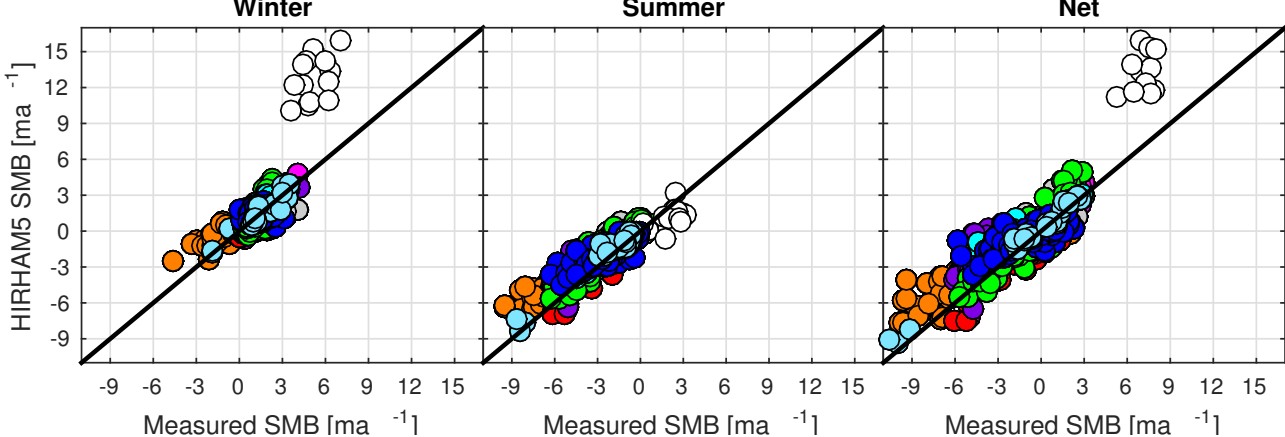

**Figure 7.** Comparison of SMB measurements from Vatnajökull ice cap from 1995-2014 and HIRHAM5 simulated values. Different colors represent different outlet glaciers, see Fig. 1. The white dots are from a point on Öræfajökull.





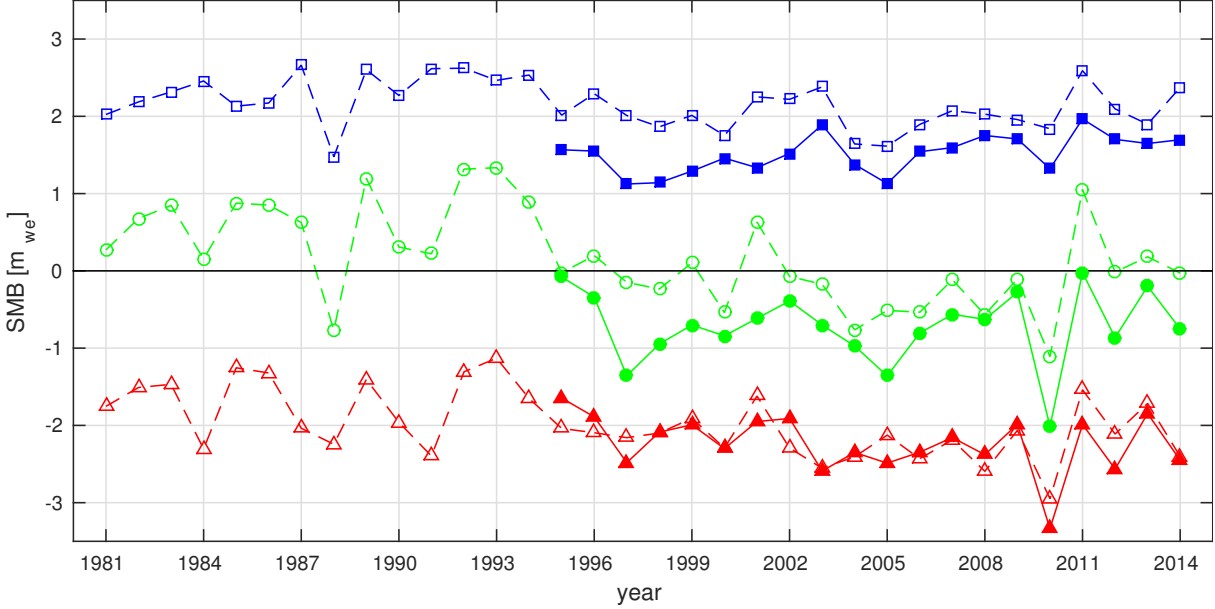

**Figure 8.** Average summer (red lines), winter (blue lines) and net (green lines) specific surface mass balance for the whole of Vatnajökull. The solid lines are the mass balance of Vatnajökull based on mass balance measurements and manual interpolation, while the dashed lines are the mass balance as simulated by HIRHAM5.





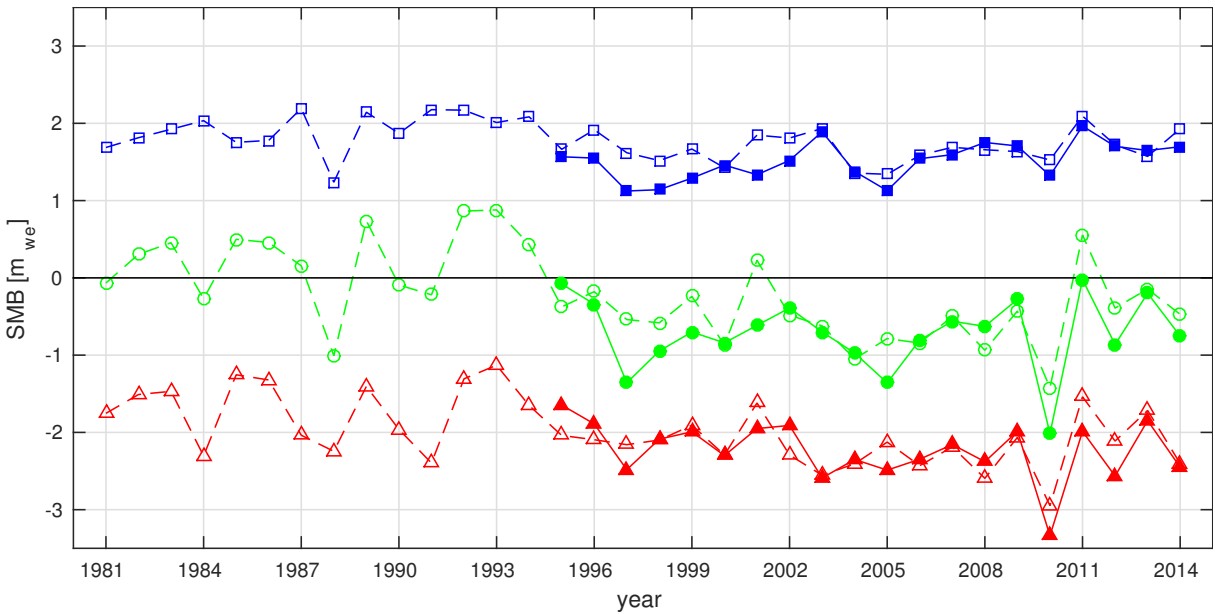

**Figure 9.** Same as Fig. 8, but corrected at the Öræfajökull area by reducing the HIRHAM5 simulated winter balance with 50%.





**Figure 10.** The average (a) winter, (b) summer, and (c) net SMB simulated by HIRHAM5 from the 1980-81 glaciological year to 2013-14. The contour lines marks the approximate placement of the ELA, which generally lies between approximately 1100 and 1300 m elevation. Figures (d)-(f) show the average deviation between model and observations over the observation period for each measurement location for the (d) winter, (e) summer, and (f) whole glaciological year.



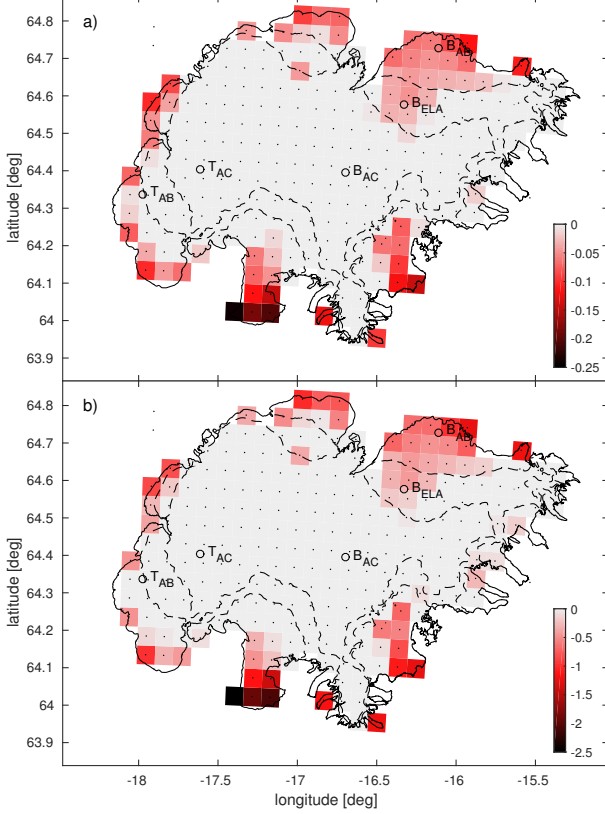

**Figure 11.** Difference between two runs with HIRHAM5, one using a MODIS bare ice albedo map and the other with a constant ice albedo in a) mean albedo and b) mean annual mass balance in m w.eq. from 2001-2014. The locations of the used AWS are shown with black circles.