# Peer review of "The importance of accurate glacier albedo for estimates of surface mass balance on Vatnajökull: Evaluating the surface energy budget in a Regional Climate Model with automatic weather station observations"

_The Cryosphere, 2017_

## Referee Comment (RC1) · Anonymous Referee #1 · 28 Mar 2017

**Review of: "The importance of accurate glacier albedo for estimates of surface mass balance on Vatnajökull: Evaluating the surface energy budget in a Regional Climate Model with automatic weather station observations ",** by *L. Steffensen Schmidt et al.*, submitted to *The Cryosphere*.

**General comments**

This manuscript is a sound evaluation of the surface mass balance (SMB) and energy balance (SEB) simulated by the regional climate model HIRHAM5 over Vatnajökull ice cap, Iceland. Here HIRHAM5 is run at 5.5 km resolution for the period 1981-2014, using an updated albedo scheme that calculates snow albedo as a function of surface temperature and snow ageing, and prescribes ice albedo from MODIS records. Comparison of HIRHAM5 output with SMB measurements (1995-2014), meteorological data, and observed radiative and turbulent heat fluxes (2001-2014) collected at 5 automatic weather stations (AWS) shows good agreement. However, the authors find a winter mass balance overestimation in the ablation zone, resulting from overestimated surface albedo in HIRHAM5. This is attributed to both the formation of a too thick snow layer covering the ice in winter and the fact that snow darkening from dust events or volcanic eruptions is not accounted for in the model.

This study investigates the climate of an Icelandic ice cap, for which little research has been conducted. Through model evaluation, the authors highlight the importance of well representing impurities deposition, e.g. from dust and volcanic ashes, to realistically capture snow/ice albedo and hence accurately model SMB changes. They also present a 1981-2014 SMB data set that will be valuable for forthcoming studies. However, further clarifications, shortening, and copy editing are necessary to improve the manuscript readability (see **Point Comments**). I judge that **minor revisions** are required before acceptance in The Cryosphere.

**Substantive Comments**

a) The authors use multiple terminologies for surface mass balance (SMB), which is confusing. For consistency, the authors should refer to "winter or summer mass balance" and "SMB or net SMB".

b) In the abstract and conclusions, the authors introduce results that are not discussed in the main manuscript. Examples can be found at **L14-15** of page 1 and **L3-10** of page 17. As the paper focuses on model evaluation, I would advise to remove these lines.

c) Ice albedo records from AWS stations are sometimes extremely low, e.g. 0.03 (L12 of page 10) and 0.01 (L26 of page 10). Are these measurements valid, i.e. deposition of dust or ashes darkening the surface, or do they result from AWS malfunction, e.g. low solar zenith angle, riming of the sensors, …? Could the authors provide references for such low albedo records or verify that all measurements used in this study are valid?

d) In Section 4.5, the authors should describe the "total energy balance" using an equation:

E = LWnet + SWnet + Hs+l + Gh

Where LWnet and SWnet are the net short/lonwave radiation, Hs+l are the turbulent heat fluxes and Gh is the ground heat flux. I would advise to refer to "melt energy" instead of "energy balance" in the discussion.

**Point Comments**

**Page 1**
**L1:** "carried out" instead of "made".
**L2-3:** I would suggest "[...] of the glacier surface mass balance (SMB). This simulation uses a new snow albedo parameterization that describes the albedo using an exponential [...] surface temperature depend**e**nt".
**L6:** "in situ SMB measurements". See also my ***Substantive Comment*** a).
**L6:** "The model agrees well with observations at the AWS sites [...]".
**L5-6**: "for 2001-2014" and "for 199**5**-2014".
**L9:** "[...] and not taking the surface darkening from dirt and [...]".
**L10-14:** "balance for the whole of Vatnajökull (1995-2014) [...], with a small mass balance underestimation of [...] on average, whereas the winter mass balance is overestimated by 0.5 m w. eq. due to too large precipitation [...] the ice cap. A simple correction [...].".
**L14-18:** I would reformulate as "Here, we use HIRHAM5 to simulate the evolution of the SMB of Vatnajökull for the period 1981-2014, and show the importance [...] ice albedo to model realistic SMB and that processes such as dust storms, currently not accounted for in RCMs, are an important [...].". See also my ***Substantive Comment*** b).

**Page 2**
**L5:** "contribute to rise the sea level by 1 cm".
**L6:** You should move the following sentence here "Runoff from Vatna. ice cap is economically important for hydropower [...] and future surface mass balance (SMB) changes are thus of keen interest.".
**L9:** "However, to carry out reliable future projections, or reconstruct the past climate, it is important to evaluate how well models simulate the present climate.".
**L11-14:** You could also refer to the work of Fettweis et al. 2017 (The Cryosphere Discussion) after Langen et al. 2016 at L13.
**L16-22:** I would suggest "Therefore, Icelandic glaciers are excellent candidates for evaluating modelled meteorological and SMB components. Compared to Greenland, observations are recorded in a relatively small area, offering a good [...] HIRHAM5 model on a regional scale. As albedo in Iceland is significantly different from that of [...], model evaluation over Iceland provides important [...] on the glacier energy balance."
**L23-26:** I would suggest: "Here we present a 1981-2014 SMB data set of the Vatna. ice cap modelled by HIRHAM5 at 5.5 km resolution. HIRHAM5 is a state-of-the-art, high resolution RCM that has been well validated over Greenland (e.g. ...). In this study, HIRHAM5 incorporates an updated albedo scheme, using a background MODIS ice albedo field, in the aim of capturing the effect of dust and tephra on ice albedo in the ablation zone. Model simulations results [...]"
**L30:** This sentence can be removed.

**Page 3**
**L5:** Could you mention the period of observation in brackets?
**L13:** I would suggest: "The turbulent fluxes, combining sensible and latent heat fluxes, and [...].".
**L25:** "weigh**t**ing"
**L27:** Remove "the" before 1995.
**L30-:** As the MODIS ice albedo product is described in this Section, the authors should move **L14-22** of page 6 here. The authors do not mention the period over which minimum autumn MODIS albedo is averaged nor the range of values obtained. This should be clarified.
**L30:** Replace "domain" by "spectrum".
**L32:** Replace "have been shown t be" by "are".

**Page 4**
**L6:** Replace "has implemented" by "implements".
**L18:** Replace "calculated results" by "calculated turbulent fluxes".

**Page 5**
**L18:** For consistency, I would suggest to refer to "dry regime" instead "cold regime", to match the regime names at **L25**.
**L19:** "In a dry regime, [...]".

**L30:** "Refreshment of the snow albedo to its minimum value […]. A partial refreshment is possible as the snow albedo is only reset to the […]".

**Page 6**
**L1:** Replace "value" by "threshold".
**L9:** I would suggest: "In the case of shallow snow cover, […]".
**L18:** I suggest: "Additional tephra or dust deposition will […]".
**L19:** Washed off by runoff or wind? Could you provide a reference here?
**L26:** Move "(equivalent to ~5.5 km)" after "0.05º". Insert "for the period 1981-2014" after "rotated pole grid".

**Page 7**
**L9:** Could you provide a reference in which this previous HIRHAM5 data set is used?
**L11:** You could remove the sentence "Running the model […] cost of the model".
**L12:** You should insert **L19-24** here.
**L14:** "effect on upward short and longwave radiation ".
**L30:** What do the authors mean by "four surrounding", do they mean the four closest grid-cells?
**L32:** For consistency, temperature should be expressed in ºC.
**L32:** "Pressure is corrected using Eq. 1 decreasing the bias down to 0.1 to 0.5 hPa".
**L33:** Replace "[…] , it is not large […]" by "[…], and the resulting differences are not large […]".

**Page 8**
**L12:** Replace "made by AWSs" by "collected at AWSs".
**L14:** Do you mean "bi-linearly interpolating"?
**L16:** Replace "given in this study" by "listed in Tables 2-4".

**Section 4.1:** Here you could include scatterplots of the 4 meteorological variables to highlight how HIRHAM5 performs on a daily basis.
**L20-21:** You could remove the sentence "Before validating […] are simulated in the model.".
**L22:** "2 m temperature".
**L25:** I suggest "The comparison of modelled and observed mean daily […] from 2001-2014 is shown in Table 2.".
**L27:** You could remove "is generally forecast with a high degree of skill;"
**L27-29:** I would suggest "At each station […] correlation (r > 0.9) between modelled and estimated pressure (Eq. 1), for the entire time series and for each individual year.".
**L31:** "by 0.8 ºC overall.".

**Page 9**
**L1:** Replace "remaining" by "other" and "but with less than 0.6 K" by "by at most 0.6 ºC".
**L2:** Insert "(r ~ 0.9)" after "all five stations".
**L17-21:** I would suggest "As a result, a similar underestimation of incoming longwave radiation is obtained at all five stations, with the largest difference occurring at the BAB station (Fig. 2). The average percentage […] (see Table 3), and falls well within the 10 % […]. However, Fig. 2a also shows that 25-30 % of the simulated days have larger errors than 10 %.".
**L28:** "[…] reproduces the daily values well (r ~ …).".
**L30:** Replace "and only" by "combined with".
**L31:** I would suggest "[…] at all AWS locations (-7.9Wm-2)."

**Page 10**
**L10-11:** I suggest: "[…] in the model, while snow cover persists longer in reality. One exception occurs in 2001, where the modelled albedo never drops down to the ice value, whereas observations […]".
**L14:** Which period? I also suggest: "which contributes to delay the albedo drop […]".
**L15:** "[…] a too thick snow cover in winter is also the cause […]".
**L15:** You could move **L20-23** here, followed by "As a result, the ice surface is never exposed […] any of the modelled years […] during all but two years, i.e. … and … . During these two years, the simulated albedo fits well […]".
**L19-20:** You could remove these sentences.

**L20:** Comparisons with mass balance […] at this station. An overestimation of the snow thickness […] fluxes, lead to persistent snow cover at the end of summer.".
**L26:** See my *Substantive Comment* c).
**L29-30:** I suggest "Close to the equilibrium line, the albedo is highly […] spatially, e.g. there is a large […]".
**L33:** "meaning that".

**Page 11**
**L3:** I suggest "The smallest difference between modelled and observed albedo is found […]".
**L5:** "An exception to this is found in 2010 […]".
**L8:** I would suggest "For instance, the very low albedo values obtained at the TAC station (Fig. 3b) are due to tephra deposition […]".
**L12:** I suggest "Such discrepancy could be explained by dust events, advancing or delaying the drop in surface albedo. Dragosics et al. (2016) investigated […]".
**L15:** "[…] all events and showed that the dust storms have a […]".
**L16:** Remove ", of course,".
**L21-25:** I would suggest "As both the incoming and outgoing SW radiation are underestimated at most stations, the net SW shows a negative bias of ∼ -6 to 12 Wm-2 at stations AB and ELA, and of -22 and -28 Wm-2 at the two AC stations. The resulting average model error at all five stations is -15.5 Wm-2.".
**L27:** I suggest "As HIRHAM5 underestimates meteorological variables at all stations, similar underestimation is obtained for the turbulent fluxes (Table 3 and Fig. 4). The two AC stations […] between the AWS estimate and […].".

**Page 12: See my *Substantive Comment* d)**
**L5:** "inaccurate cloud representation cannot be the only […] error. Errors in the interaction of clouds and radiation, e.g. error in the optical thickness of the clouds, or in the clear sky fluxes, could partly explain these discrepancies.".
**L10:** I suggest "Since the simulated outgoing […] a small negative bias, the deviation in net LW radiation is governed by the incoming radiation. Errors in the simulated albedo mean […] the deviation in net SW radiation. These errors can be partly attributed to […] storms, which are not taken into account in HIRHAM5. In addition, errors in the simulated albedo also stem from snow cover that disappears too slowly compared to AWS records in the ablation zone. As a result, modelled albedo drops […]".
**L16:** I suggest "of the net SW and LW radiation and the turbulent fluxes leads to underestimated melt energy, which contributes to overestimate the modelled snow thickness.".
**L21:** "the mean difference between modelled and observed energy components […] is shown for each station (Fig. 5)".
**L25:** "net SW radiation".
**L26-28:** These explanations are unclear to me, could you reformulate?
**L32:** "Modelled longwave radiation is consistently underestimated by 10 Wm-2.".
**L34:** "the albedo comparison. Depending on […] the albedo is generally […]".

**Page 13**
**L6:** "As previously discussed, this albedo bias, and hence underestimated SW radiation, occurs […] proximity of the equilibrium line. An underestimation of the incoming […]".
**L19:** Replace "offers an evaluation" by "allows to evaluate".
**L20:** Here the authors could mention the specific year.

**Page 14**
**L2:** "SMB is also measured at 25-120 non-AWS sites, depending on the year (Fig. 1b).".
**L8:** "receives a large amount of precipitation. However, since HIRHAM5 […]".
**L13:** Replace "particularly significant process" by "key process".
**L15:** I would suggest "Removing this location from the comparison, the total difference drops to one-third […]".
**L16-18:** This sentence is difficult to read, could you reformulate?
**L26:** I would suggest "HIRHAM is used to estimate the mean SMB of Vatna. for 1981-2014. The winter, summer and net mass balances […]."
**L29:** "manually interpolating", what do the authors mean by this? Please clarify.

**L31:** "The largest deviations are obtained in 1995, where ablation is overestimated in the simulation [...] 2010-2012, where ablation is underestimated [...]".

**Page 15**
**L16-18:** I would advise to swap Fig. 10 and Fig. 9, and to discuss these mass balance maps earlier in this Section.
**L21:** "a previous run using a constant ice albedo of 0.3.".
**L26:** Replace "appears to be on the" by "are found on the".
**L27:** "are located in areas".
**L28:** "The TAB station is located in the ablation area, where the ice surface is never exposed in the model due to an overestimation of the winter accumulation.".

**Page 16:** Present tense should be used in the conclusions.
**L1:** "[...] ice cap allows us to evaluate the model performance.".
**L21:** "[...] into the model is to implement a stochastic [...]".
**L31:** "by 0.06 m".

**Page 17**
**L3-10:** See my **_Substantive Comment_** a).
**L10:** "HIRHAM5 is therefore a useful tool to expand [...]".
**L15:** "[...] lateral boundary, e.g. output of a general circulation model.".

**Figures and Tables**

**Tables 2-4:** I would advise to show average observations at the AWS stations instead of HIRHAM values in the second column.
**Table 2:** For consistency, temperature should be expressed in ºC.
**Figure 1a:** Could you rename the different stations so that they match the labels used in the main manuscript, e.g. T acc. → T-AC.
**Figures 2-4:** Could you use similar symbols for both locations (B and T stations), the large crosses you use make the deviations appear larger than they really are.
**Figure 4:** Remove the last sentence in the caption and insert a similar legend (symbols) as in Figs. 2 and 3
**Figure 5:** In the legend, could you write "Hs+l" instead of "Hs + Hl"?
**Figure 6 caption:** I suggest: "Comparison of the winter [...] 2014 between the mass [...]".
**Figure 7 caption:** "see Fig. 1b".
**Figure 10 caption:** Replace "placement" by "location".
**Figure 11 caption:** "Difference in a) mean albedo, and b) mean SMB in m w. eq. for 2001-2014 between two runs with [...].".

---

## Referee Comment (RC2) · Anonymous Referee #2 · 26 Apr 2017

**The importance of accurate glacier albedo for estimates of surface mass balance on Vatnajökull: Evaluating the surface energy budget in a Regional Climate Model with automatic weather station observations**

Louise Steffensen Schmidt, Guðfinna Aðalgeirsdóttir, Sverrir Guðmundsson, Peter L. Langen, Finnur Pálsson, Ruth Mottram, Simon Gascoin, and Helgi Björnsson

**Summary**:
The authors present a simulation of mass balance for the Vatnajökull ice cap using the HIRHAM5 regional climate model, with an updated albedo scheme that simulates albedo as a function of snow age and surface temperature. The simulated mass and energy balance are compared with observations from automatic weather stations on the ice cap. There is a fairly good agreement between observed and measured mass and energy balance, with the largest differences being associated with errors in simulated albedo. These errors are associated with inaccuracies in simulating snow cover extent during summer, as well as the lack of a scheme for accounting for impurity deposition in the model.

**General Comments:**
The paper is well written, well thought out, and scientifically sound. The paper is an important contribution as it focuses on regional climate model simulation of albedo over an ice cap and identifies challenges that can be addressed by future work. I believe the paper should be accepted for publication in the Cryosphere after relatively minor revisions discussed below. The points below are mostly very minor changes.

Some general points are:

1. Since a main focus of the paper is on albedo and how it influences mass balance, some papers discussing the importance of albedo to glacier and ice sheet mass balance and challenges in modeling albedo should be mentioned in the introduction.
2. Though this is not essential, I feel that the methods section could benefit by being reorganized. Since the main focus of the paper is validating the regional model results, the RCM could be described first, followed by the description of observational data, followed by the description of methods of comparison (including AWS point models – section 3.1, validation methods 3.2.2, and elevation-based corrections 3.2.5). This would require some editing to ensure that the text is consistent with the new order.
3. Figure 10 is hardly discussed in section 4.7. There should be more discussion of this figure. In particular, the model – measured differences for the weather station measurements are consistent with the differences shown in Figs. 10 d, e, and f; for example there is a low SMB bias at high elevations and high SMB bias at low elevations. These consistencies should be discussed.
4. Section 4.8 also seems very short. The authors could provide more discussion of how the albedo differences affect SMB, and how this relates to the biases discussed in other parts of the study.

**Specific Comments:**

1. **P. 1, Line 14:** Suggest changing "specific mass balance" to "specific surface mass balance" for clarity.

2. **P. 1, Line 16:** Add "through 2014" after "from 1981" to make the time period clear.

3. **P. 1, Lines 16-18:** The second part of the sentence doesn't fit with the first part, and contradicts it somewhat. I think the point the authors are trying to make here is that the model can provide a reasonable representation of surface mass balance, but that a major source of uncertainty in this representation is the representation of surface albedo and how it evolves. Please clarify.

4. **P. 2, Line 18:** "Good records" is a bit vague. What is good about them?

5. **P. 2, Line 25:** Change "background albedo" to "background bare ice albedo" for clarity.

6. **P. 2, Lines 24-26:** I believe van Angelen et al. (2012) was the first to use this approach. This paper should be cited:
   van Angelen, J. H., Lenaerts, J. T. M., Lhermitte, S., Fettweis X., Kuipers Munneke, P., van den Broeke, M. R., van Meijgaad, and Smeets, C. J. P. P.: Sensitivity of Greenland Ice Sheet surface mass balance to surface albedo parameterization: a study with a regional climate model, The Cryosphere, 6, 1175-1186, doi: 10.5194/tc-6-1175-2012, 2012.

7. **P. 3, Line 6:** Note that Brúarjökull and Tungnaárjökull are glaciers that make up part of the Vatnajökull ice cap.

8. **P. 3, Line 25:** How is the summer surface identified?

9. **P. 3, Line 30:** Which MODIS product is used here?

10. **P. 4, Line 5:** add "from AWS measurements" after "The turbulent energy fluxes were calculated" for clarity.

11. **P. 6, Lines 20-21:** How is it known that the new particles are generally washed off? Isn't it possible that some of the impurities are scavenged at the surface? (e.g. Doherty et al., 2013)
    Doherty, S. J., Grenfell, T. C., Forsström, S., Hegg, D. L., Brandt, R. E., and Warren, S. G.: Observed vertical distribution of black carbon and other insoluble light-absorbing particles in melting snow, J. Geophys. Res., 118, 1-17, doi: 10.1002/jgrd.50235, 2013.

12. **P. 7, Line 31:** I suggest noting here that the correction was applied so that model results could be compared to AWS measurements at AWS locations.

13. **P. 8, Lines 14-15:** This repeats some information from section 3.2.3. Since the corrections made in section 3.2.3 are done for the purpose of validation, perhaps the material from section 3.2.3 can be merged into this section.

14. **P. 8, Line 16:** It is unclear what "components" refers to here.

15. **P. 8, Line 22:** Change "temperature, $T_{2m}$" to "air temperature at 2 m, $T_{2m}$" for clarity.

16. **P. 9, Line 7:** What is the temperature gradient between?

17. **P. 9, Lines 12-14:** Can the author's elaborate briefly on this? Why are the winds interpolated rather than being calculated within the model?

18. **P. 9, Line 31:** Suggest changing "total LW" to "net LW (incoming-outgoing)" radiation

19. **P. 10, Line 10:** By "generally exposed" do the authors mean "every year"?

20. **P. 10, Line 19:** Since the difficulties in modeling the ELA station have not been elaborated on yet, perhaps the difficulties should be briefly summarized, e.g. "some of the modelling difficulties which affect the ELA station (discussed below), associated with errors in simulating the presence or absence of snow cover…"

21. **P. 11, Line 7:** I believe "underestimating the albedo" should be changed to "overestimating the albedo".

22. **P. 12, Line 2:** Suggest changing "total energy was estimated" to "total energy balance was estimated".

23. **P. 13, Line 1:** "the summer surface was reached" is a bit unclear. Are the authors referring to exposure of bare ice at this location?

24. **P. 13, Line 5:** Change "SW radiation" to "net SW radiation" for clarity.

25. **P. 13, Line 8:** Again "net SW radiation" would be clearer.

26. **P. 16, Line 4:** Be more clear about what is underestimated.

27. **P. 16, Line 12:** It is known that the model simulates surface temperatures well, as discussed in the previous paragraph. Perhaps it is better to say that the accuracy of outgoing longwave radiation is consistent with the ability of the model to capture surface temperatures.

28. **P. 16, Lines 15-16:** The better agreement with observations as compared with a fixed albedo, though obvious given the wide spread of observed values, is not mentioned in the results section. If mentioned here, it should also be mentioned in Section 4.3.

29. **P. 16, Lines 30-31:** This sentence is confusing. It makes it seem as if the average modeled mass balance for 1981-2014 is being compared with the average for 1995-2014 from observations. Rather, the model results for 1995-2014 were compared with observations for 1995-2014. Please clarify.

30. **Table 2:** In the caption, the meaning of the parameters in column 1 should be explained, as is done for Table 3.

31. **Figure 1:** The weather station names are not consistent with the names in the text. For example "B-abl" should be "B-AB" to be consistent with the text. Also, in the caption, it should be pointed out that the unlabeled sites in Fig. 1a were not used in the study. Optionally, the symbols could be a different color to emphasize this. Perhaps Brúarjökull and Tungnaárjokull could also be labeled on the map for clarity. The lines on Fig. 1b are not explained. I suppose these connect mass balance sites collected along a transect. Finally, the labels (a) and (b) should be added for the sub-plots.

32. **Figure 10:** (Caption) Add the years of the observational period for clarity.

**Technical Corrections:**

1. **P. 1, Line 3:** Suggest changing "describes the albedo with an exponential decay with time…" to "allows albedo to exponentially decay with time…"

2. **P. 4, Line 1:** Change "lat/lon" to "latitude-longitude coordinates"

3. **P. 5, Line 28:** Change "The found best-fit values were…" to "The best-fit values were found to be…"

4. **P. 5, Line 30:** Change "Refreshment of albedo to the maximum value only occurs…" to "Albedo is only refreshed to the maximum value if.."

5. **P. 6, Line 11:** I believe there is a typo in the equation. Should "$d^{n+1}$" be "$d^{t+1}$"?

6. **P. 6, Line 23:** Change "How much this" to "The extent to which"

7. **P. 7, Line 4:** Change "The model is here run" to "For this study, the model is run"

   The sentence has been deleted and the period added to the first line after suggestion from referee #1

8. **P. 7, Line 10:** Change "allows a quick and thorough" to "allows for a quick and thorough"

9. **P. 8, Line 5:** Change "like for example that of the albedo" to "including, for example, the albedo parameterization,"

10. **P. 8, Line 31:** Change "with 0.8 K overall" to "by 0.8 K on average"

11. **P. 9, Line 1:** Change "but with less than 0.6 K" to "by less than 0.6 K";  change "it for example" to "for example, it"
12. **P. 9, Line 20:** Change "larger errors-" to "larger errors;"
13. **P. 10, Line 12:**  Change "down to 0.03" to "as low as 0.03"; Change "the total overestimation" to "the average overestimation"
14. **P. 10, Line 27:** Change "that low in situ…" to "that a low in situ…".
15. **P. 11, Line 28:** Change "comparisons statistics" to "comparison statistics".
16. **P. 12, Lines 10-12:** This sentence is rather long.  I suggest splitting it into two sentences.
17. **P. 14, Line 16:** Change "one-third that of the AWS sites…" to "one-third the difference with respect to the AWS sites…"
18. **P. 14, Line 27:** Change "back to 1981" to "extending back to 1981".
19. **P. 16, Line 6:** Change "comparisons only uses" to "comparisons only use".
20. **P. 16, Line 12:** Change "there was a larger differences" to "there was a larger difference".
21. **P. 16, Line 19:** Change "and that the model does not account" to "and the fact that the model does not account"
22. **P. 16, Line 20:** Change "way to include" to "means of capturing"
23. **P. 17, Line 21:** Change "could be including a stochastic…" to "could be to include a stochastic…"
24. **P. 17, Line 31:** Change "with 0.06 m" to "by 0.06 m".
25. **P. 18, Line 13:** Change "like for example ERA-20C" to "for example, with the ERA20C reanalysis".
26. **Figure 3:** The axis for Fig. 3b is a bit confusing.  I suggest removing the 100, and leaving 0 for all plots.
27. **Figure 4:** suggest adding "from AWS stations" after "fluxes calculated" for clarity.
28. **Figure 6:** The caption seems to be erroneously in italics.
29. **Figure 11:** Change "used AWS" to "AWS stations used in this study"

---

## Author Comment (AC1) · 24 May 2017

**Response to review #1**

We would first like to thank the reviewer for his useful and detailed comments which have helped a lot to improve the readability of our manuscript.

**General comments**

This manuscript is a sound evaluation of the surface mass balance (SMB) and energy balance (SEB) simulated by the regional climate model HIRHAM5 over Vatnajökull ice cap, Iceland. Here HIRHAM5 is run at 5.5 km resolution for the period 1981-2014, using an updated albedo scheme that calculates snow albedo as a function of surface temperature and snow ageing, and prescribes ice albedo from MODIS records. Comparison of HIRHAM5 output with SMB measurements (1995-2014), meteorological data, and observed radiative and turbulent heat fluxes (2001-2014) collected at 5 automatic weather stations (AWS) shows good agreement. However, the authors find a winter mass balance overestimation in the ablation zone, resulting from overestimated surface albedo in HIRHAM5. This is attributed to both the formation of a too thick snow layer covering the ice in winter and the fact that snow darkening from dust events or volcanic eruptions is not accounted for in the model.

This study investigates the climate of an Icelandic ice cap, for which little research has been conducted.

Actually, much research has been conducted on this ice cap, which is part of what makes this study possible. For example, mass balance measurements have been conducted since 1991-92 glaciological year and weather stations have been operated on the glacier since 1994. However, it is correct that not much research has been done on this ice cap using a Regional Climate model.

Through model evaluation, the authors highlight the importance of well representing impurities deposition, e.g. from dust and volcanic ashes, to realistically capture snow/ice albedo and hence accurately model SMB changes. They also present a 1981-2014 SMB data set that will be valuable for forthcoming studies. However, further clarifications, shortening, and copy editing are necessary to improve the manuscript readability (see *Point Comments*). I judge that **minor revisions** are required before acceptance in The Cryosphere.

**Substantive Comments**

a) The authors use multiple terminologies for surface mass balance (SMB), which is confusing. For consistency, the authors should refer to "winter or summer mass balance" and "SMB or net SMB".

   Thanks for bringing this to our attention. We have changed the manuscript so we only use one terminology.

b) In the abstract and conclusions, the authors introduce results that are not discussed in the main manuscript. Examples can be found at **L14-15** of page 1 and **L3-10** of page 17. As the paper focuses on model evaluation, I would advise to remove these lines.

   We think this is an important point to get across, but you are right that it perhaps does not belong in the abstract and conclusion. We have removed the lines from the abstract

and added L3-10 in the conclusion to section 4.7 instead, so it is still included in the paper but not as prominently.

c) Ice albedo records from AWS stations are sometimes extremely low, e.g. 0.03 (L12 of page 10) and 0.01 (L26 of page 10). Are these measurements valid, i.e. deposition of dust or ashes darkening the surface, or do they result from AWS malfunction, e.g. low solar zenith angle, riming of the sensors, …? Could the authors provide references for such low albedo records or verify that all measurements used in this study are valid?

Very low values of the ice albedo have regularly been observed in the ablation zone in Vatnajökull, down to values of 0.05. These low values have also been observed in MODIS measurements, with the MCD43A MODIS product e.g. observing values at Brúarjökull down to 0.03. In the case of extremely low values (lower than 0.05), there are some years where the stations have been placed on a layer of tephra or sand, and therefore the very low albedo values may not represent the albedo of the ice but more likely the albedo of the tephra. A sentence to this effect has been added to the paper (after the 0.03 mention).

d) In Section 4.5, the authors should describe the "total energy balance" using an equation:

E = LWnet + SWnet + Hs+l + Gh

Where LWnet and SWnet are the net short/lonwave radiation, Hs+l are the turbulent heat fluxes and Gh is the ground heat flux. I would advise to refer to "melt energy" instead of "energy balance" in the discussion.

The surface energy balance equation has been added to the section

**Point Comments**

**Page 1**
**L1:** "carried out" instead of "made".
**L2-3:** I would suggest "[…] of the glacier surface mass balance (SMB). This simulation uses a new snow albedo parameterization that describes the albedo using an exponential […] surface temperature dependent".
**L6:** "in situ SMB measurements". See also my **Substantive Comment** a).
**L6:** "The model agrees well with observations at the AWS sites […]".
**L5-6**: "for 2001-2014" and "for 1995-2014".
**L9:** "[…] and not taking the surface darkening from dirt and […]".
**L10-14:** "balance for the whole of Vatnajökull (1995-2014) […], with a small mass balance underestimation of […] on average, whereas the winter mass balance is overestimated by 0.5 m w. eq. due to too large precipitation […] the ice cap. A simple correction […].".
**L14-18:** I would reformulate as "Here, we use HIRHAM5 to simulate the evolution of the SMB of
Vatnajökull for the period 1981-2014, and show the importance […] ice albedo to model realistic SMB and that processes such as dust storms, currently not accounted for in RCMs, are an important […].". See also my **Substantive Comment** b).

Thanks for the suggestions. These have all been changed.

**Page 2**
**L5:** "contribute to rise the sea level by 1 cm".
**L6:** You should move the following sentence here "Runoff from Vatna. ice cap is economically important for hydropower […] and future surface mass balance (SMB) changes are thus of keen interest.".
**L9:** "However, to carry out reliable future projections, or reconstruct the past climate, it is
important to evaluate how well models simulate the present climate.".
**L11-14:** You could also refer to the work of Fettweis et al. 2017 (The Cryosphere Discussion)
after Langen et al. 2016 at L13.
**L16-22:** I would suggest "Therefore, Icelandic glaciers are excellent candidates for evaluating modelled meteorological and SMB components. Compared to Greenland, observations are recorded in a relatively small area, offering a good […] HIRHAM5 model on a regional scale. As albedo in Iceland is significantly different from that of […], model evaluation over Iceland provides important […] on the glacier energy balance."
**L23-26:** I would suggest: "Here we present a 1981-2014 SMB data set of the Vatna. ice cap modelled by HIRHAM5 at 5.5 km resolution. HIRHAM5 is a state-of-the-art, high resolution RCM that has been well validated over Greenland (e.g. …). In this study, HIRHAM5 incorporates an updated albedo scheme, using a background MODIS ice albedo field, in the aim of capturing the effect of dust and tephra on ice albedo in the ablation zone. Model simulations results […]"
**L30:** This sentence can be removed.

Done.

**Page 3**
**L5:** Could you mention the period of observation in brackets?

Of course. It's been added

**L13:** I would suggest: "The turbulent fluxes, combining sensible and latent heat fluxes, and […].".
**L25:** "weigh**t**ing"
**L27:** Remove "the" before 1995.

Thanks, it has been corrected

**L30-:** As the MODIS ice albedo product is described in this Section, the authors should move **L14-**
**22** of page 6 here. The authors do not mention the period over which minimum autumn MODIS
albedo is averaged nor the range of values obtained. This should be clarified.

The lines have been moved to the observations section. The range of values (0.03-0.3) and the period used has been added to the text.

**L30:** Replace "domain" by "spectrum".
**L32:** Replace "have been shown t be" by "are".

Done. Thanks.

**Page 4**
**L6:** Replace "has implemented" by "implements".
**L18:** Replace "calculated results" by "calculated turbulent fluxes".

**Page 5**
**L18:** For consistency, I would suggest to refer to "dry regime" instead "cold regime", to match the
regime names at **L25**.
**L19:** "In a dry regime, […]"

**L30:** "Refreshment of the snow albedo to its minimum value […]. A partial refreshment is
possible as the snow albedo is only reset to the […]".

**Page 6**
**L1:** Replace "value" by "threshold".
**L9:** I would suggest: "In the case of shallow snow cover, […]".
**L18:** I suggest: "Additional tephra or dust deposition will […]".

Okay, thanks. All suggestions have been added

**L19:** Washed off by runoff or wind? Could you provide a reference here?

Washed off by runoff. This has not been published, but it has been observed during field visits to e.g. Langjökull and Brúarjökull over the summer during the last 20+ years, but of course there is a chance that some of the particles remain. However, these have a small effect compared to the tephra layers. That this is based on field observations has been clarified in the text.

**L26:** Move "(equivalent to ~5.5 km)" after "0.05º". Insert "for the period 1981-2014" after "rotated pole grid".

Done

**Page 7**
**L9:** Could you provide a reference in which this previous HIRHAM5 data set is used?

Yes, I've added a reference to Langen et al, 2017, which also use this HIRHAM5 data set

**L11:** You could remove the sentence "Running the model […] cost of the model".
**L12:** You should insert **L19-24** here.

Done

**L14:** "effect on upward short and longwave radiation ".

The albedo scheme will have an effect on the upward shortwave radiation, so the sentence remains "effect on upward longwave radiation"

**L30:** What do the authors mean by "four surrounding", do they mean the four closest grid-cells?

Yes, we do. It has been changed.

**L32:** For consistency, temperature should be expressed in ºC.

We would prefer to keep the temperature in SI units. Previous mentions of temperature in ºC have instead been changed to Kelvin.

**L32:** "Pressure is corrected using Eq. 1 decreasing the bias down to 0.1 to 0.5 hPa".
**L33:** Replace "[…], it is not large […]" by "[…], and the resulting differences are not large […]".

**Page 8**
**L12:** Replace "made by AWSs" by "collected at AWSs".

Done

**L14:** Do you mean "bi-linearly interpolating"?

Yes, we do

**L16:** Replace "given in this study" by "listed in Tables 2-4".

Ok, done

**Section 4.1:** Here you could include scatterplots of the 4 meteorological variables to highlight
how HIRHAM5 performs on a daily basis.

Sure. The figure has been added as Fig. 2

**L20-21:** You could remove the sentence "Before validating […] are simulated in the model.".
**L22:** "2 m temperature".
**L25:** I suggest "The comparison of modelled and observed mean daily […] from 2001-2014 is
shown in Table 2.".
**L27:** You could remove "is generally forecast with a high degree of skill;"
**L27-29:** I would suggest "At each station […] correlation (r > 0.9) between modelled and
estimated pressure (Eq. 1), for the entire time series and for each individual year.".
**L31:** "by 0.8 ºC overall."

Corrected, except we decided to use K instead of ºC in L31.

**Page 9**
**L1:** Replace "remaining" by "other" and "but with less than 0.6 K" by "by at most 0.6 ºC".
**L2:** Insert "(r ~ 0.9)" after "all five stations".
**L17-21:** I would suggest "As a result, a similar underestimation of incoming longwave radiation is obtained at all five stations, with the largest difference occurring at the BAB station (Fig. 2). The average percentage […] (see Table 3), and falls well within the 10 % […]. However, Fig. 2a also shows that 25-30 % of the simulated days have larger errors than 10 %.".
**L28:** "[…] reproduces the daily values well (r ~ …).".
**L30:** Replace "and only" by "combined with".
**L31:** I would suggest "[…] at all AWS locations (-7.9Wm-2)."

**Page 10**
**L10-11:** I suggest: "[…] in the model, while snow cover persists longer in reality. One exception occurs in 2001, where the modelled albedo never drops down to the ice value, whereas observations […]".

Thanks, these have all been changed

**L14:** Which period? I also suggest: "which contributes to delay the albedo drop […]".

The measurement period (2001-2014). And the suggestion has been added.

**L15:** "[…] a too thick snow cover in winter is also the cause […]".
**L15:** You could move **L20-23** here, followed by "As a result, the ice surface is never exposed […] any of the modelled years […] during all but two years, i.e. … and … . During these two years, the simulated albedo fits well […]".
**L19-20:** You could remove these sentences.

**L20:** Comparisons with mass balance […] at this station. An overestimation of the snow thickness […] fluxes, lead to persistent snow cover at the end of summer.".

Corrected. Thanks

**L26:** See my *Substantive Comment* c).

See answer under Substantive Comments c)

**L29-30:** I suggest "Close to the equilibrium line, the albedo is highly […] spatially, e.g. there is a
large […]".
**L33:** "meaning that".

Done

**Page 11**
**L3:** I suggest "The smallest difference between modelled and observed albedo is found […]".
**L5:** "An exception to this is found in 2010 […]".
**L8:** I would suggest "For instance, the very low albedo values obtained at the TAC station (Fig.
3b) are due to tephra deposition […]".
**L12:** I suggest "Such discrepancy could be explained by dust events, advancing or delaying the
drop in surface albedo. Dragosics et al. (2016) investigated […]".
**L15:** "[…] all events and showed that the dust storms have a […]".
**L16:** Remove ", of course,".
**L21-25:** I would suggest "As both the incoming and outgoing SW radiation are underestimated at most stations, the net SW shows a negative bias of ~ -6 to 12 Wm-2 at stations AB and ELA, and of -22 and -28 Wm-2 at the two AC stations. The resulting average model error at all five stations is -15.5 Wm-2.".

**L27:** I suggest "As HIRHAM5 underestimates meteorological variables at all stations, similar underestimation is obtained for the turbulent fluxes (Table 3 and Fig. 4). The two AC stations […] between the AWS estimate and […].".

All suggested changes to this page have been added. Thanks

**Page 12:** See my *Substantive Comment* d)

See answer under *Substantive Comment* d)

**L5:** "inaccurate cloud representation cannot be the only […] error. Errors in the interaction of clouds and radiation, e.g. error in the optical thickness of the clouds, or in the clear sky fluxes, could partly explain these discrepancies.".
**L10:** I suggest "Since the simulated outgoing […] a small negative bias, the deviation in net LW radiation is governed by the incoming radiation. Errors in the simulated albedo mean […] the deviation in net SW radiation. These errors can be partly attributed to […] storms, which are not taken into account in HIRHAM5. In addition, errors in the simulated albedo also stem from snow cover that disappears too slowly compared to AWS records in the ablation zone. As a result, modelled albedo drops […]".
**L16:** I suggest "of the net SW and LW radiation and the turbulent fluxes leads to underestimated
melt energy, which contributes to overestimate the modelled snow thickness.".
**L21:** "the mean difference between modelled and observed energy components […] is shown for
each station (Fig. 5)".
**L25:** "net SW radiation".

Thanks for the suggestions. They have been implemented

**L26-28:** These explanations are unclear to me, could you reformulate?

We have reformulated as; *The mean difference between observations and the simulations of the SW radiation for non-eruption years is -3 W m$^{-2}$ whereas the radiation difference in 2010 is -106 W m$^{-2}$. Assuming the larger deviation from the mean in 2010 is only due to the volcanic eruption, the increase in available energy due to the eruption is 103 W m$^{-2}$. If it is further assumed that the surface was always at melting point, the increase in melt due to the 2010 Eyjafjallajökull eruption over the 128 day measuring period would be ~3.1 m w.eq. at this station.*

**L32:** "Modelled longwave radiation is consistently underestimated by 10 Wm-2.".
**L34:** "the albedo comparison. Depending on […] the albedo is generally […]".

**Page 13**
**L6:** "As previously discussed, this albedo bias, and hence underestimated SW
radiation, occurs […] proximity of the equilibrium line. An underestimation of the incoming […]".
**L19:** Replace "offers an evaluation" by "allows to evaluate".
**L20:** Here the authors could mention the specific year.

**Page 14**
**L2:** "SMB is also measured at 25-120 non-AWS sites, depending on the year (Fig. 1b).".
**L8:** "receives a large amount of precipitation. However, since HIRHAM5 […]".
**L13:** Replace "particularly significant process" by "key process".

**L15:** I would suggest "Removing this location from the comparison, the total difference drops to
one-third [...]".

*Thanks, the above changes have all been added*

**L16-18:** This sentence is difficult to read, could you reformulate?

*We have reformulated as: The reason the difference is smaller than for the AWS sites only is that more sites close to the edge of the ice cap are included. The winter balance at the measurement points in the ablation area of the icecap generally is overestimated in the model, and therefore these points partly offset the underestimation in the middle of the ice cap.*

**L26:** I would suggest "HIRHAM is used to estimate the mean SMB of Vatna. for 1981-2014. The
winter, summer and net mass balances [...].""

*changed*

**L29:** "manually interpolating", what do the authors mean by this? Please clarify

*Manually interpolated might not be the right description. They are created using Kriging interpolation of the mass balance measurements. On glaciers where no measurements are available, the mass balance is approximated using known correlations with the mass balance on other glaciers. Skerðarárjökull e.g. has no mass balance measurements, but it is known to have a similar mass balance as Breiðamerkurjökull, which is measured. The Breiðamerkurjökull balance is therefore used to estimate the Skerðarárjökull balance. The manual has been removed from the sentence, and if more info is needed about the interpolation scheme, more info is given be in the reference (Palsson, 2016).*

**L31:** "The largest deviations are obtained in 1995, where ablation is overestimated in the
simulation [...] 2010-2012, where ablation is underestimated [...]".

*changed*

**Page 15**
**L16-18:** I would advise to swap Fig. 10 and Fig. 9, and to discuss these mass balance maps earlier
in this Section.

*Done, the two figures and the sections discussing them have been switched*

**L21:** "a previous run using a constant ice albedo of 0.3.".
**L26:** Replace "appears to be on the" by "are found on the".
**L27:** "are located in areas".
**L28:** "The TAB station is located in the ablation area, where the ice surface is never exposed in
the model due to an overestimation of the winter accumulation.".

*Changed*

**Page 16:** Present tense should be used in the conclusions.
**L1:** "[...] ice cap allows us to evaluate the model performance."

**L21:** "[…] into the model is to implement a stochastic […]".
**L31:** "by 0.06 m".

changed

**Page 17**
**L3-10:** See my *Substantive Comment* a).

See comment under *Substantive Comment* a).

**L10:** "HIRHAM5 is therefore a useful tool to expand […]".
**L15:** "[…] lateral boundary, e.g. output of a general circulation model.".

Changed

**Figures and Tables**

**Tables 2-4:** I would advise to show average observations at the AWS stations instead of HIRHAM
values in the second column.

Ok, the values have been changed in the tables

**Table 2:** For consistency, temperature should be expressed in ºC.

We would prefer to use SI units. Previous uses of ºC have been changed to K for consistency.

**Figure 1a:** Could you rename the different stations so that they match the labels used in the main
manuscript, e.g. T acc. à T-AC.

Of course. This has been changed

**Figures 2-4:** Could you use similar symbols for both locations (B and T stations), the large
crosses you use make the deviations appear larger than they really are.

Definitely. Both locations now use dots in the scatter plots.

**Figure 4:** Remove the last sentence in the caption and insert a similar legend (symbols) as in Figs. 2 and 3

Done

**Figure 5:** In the legend, could you write "Hs+I" instead of "Hs + HI"?

No problem. It has been changed

**Figure 6 caption:** I suggest: "Comparison of the winter […] 2014 between the mass […]".
**Figure 7 caption:** "see Fig. 1b".
**Figure 10 caption:** Replace "placement" by "location".
**Figure 11 caption:** "Difference in a) mean albedo, and b) mean SMB in m w. eq. for 2001-2014 between two runs with […].".

All caption suggestions have been added. Thanks

---

## Author Comment (AC2) · 24 May 2017

**Response to review #2**

We would first like to thank the reviewer for his useful suggestions which have helped a lot to improve our manuscript.

**The importance of accurate glacier albedo for estimates of surface mass balance on Vatnajökull: Evaluating the surface energy budget in a Regional Climate Model with automatic weather station observations**

Louise Steffensen Schmidt, Guðfinna Aðalgeirsdóttir, Sverrir Guðmundsson, Peter L. Langen, Finnur Pálsson, Ruth Mottram, Simon Gascoin, and Helgi Björnsson

**Summary:**
The authors present a simulation of mass balance for the Vatnajökull ice cap using the HIRHAM5 regional climate model, with an updated albedo scheme that simulates albedo as a function of snow age and surface temperature. The simulated mass and energy balance are compared with observations from automatic weather stations on the ice cap. There is a fairly good agreement between observed and measured mass and energy balance, with the largest differences being associated with errors in simulated albedo. These errors are associated with inaccuracies in simulating snow cover extent during summer, as well as the lack of a scheme for accounting for impurity deposition in the model.

**General Comments:**
The paper is well written, well thought out, and scientifically sound. The paper is an important contribution as it focuses on regional climate model simulation of albedo over an ice cap and identifies challenges that can be addressed by future work. I believe the paper should be accepted for publication in the Cryosphere after relatively minor revisions discussed below. The points below are mostly very minor changes.

Some general points are:
**1.** Since a main focus of the paper is on albedo and how it influences mass balance, some papers discussing the importance of albedo to glacier and ice sheet mass balance and challenges in modeling albedo should be mentioned in the introduction.

A small section has been added about this with references to a few papers

**2.** Though this is not essential, I feel that the methods section could benefit by being reorganized. Since the main focus of the paper is validating the regional model results, the RCM could be described first, followed by the description of observational data, followed by the description of methods of comparison (including AWS point models – section 3.1, validation methods 3.2.2, and elevation-based corrections 3.2.5). This would require some editing to ensure that the text is consistent with the new order.

You're right, writing about the RCM first would be more logical. The sections have been reorganised in the manner you suggest

**3.** Figure 10 is hardly discussed in section 4.7. There should be more discussion of this figure. In particular, the model – measured differences for the weather station measurements are consistent with the

differences shown in Figs. 10 d, e, and f; for example there is a low SMB bias at high elevations and high SMB bias at low elevations. These consistencies should be discussed.

*The discussion of the figure has been expanded. Now it reads; "Spatial maps of the (uncorrected) average winter, summer, and net SMB from the 1980-81 glaciological year until 2013-14 are shown in Figure 9. The approximate location of the average ELA is marked on the figure. The model captures the position of the ELA fairly well, but at e.g. Brúarjökull, where the average ELA is at 1200 m, the position of the average ELA is at a too high elevation. The average deviation between observation and model over the observation period at each measurement location is also shown in Figure 9 in order to give an indication of the average error of the model at different parts of the ice cap. The winter balance (Fig. 9e) is generally overestimated at low elevations and underestimated at high elevations, except for at Öræfajökull where there is a large overestimation of the winter balance, as discussed in the previous section. As can be seen in Figure 9e, there is generally a low SMB bias at high elevations and a high SMB bias at low elevations during the summer. This is consistent with the comparisons with AWS stations, as we found that the bias in the energy available for melt was smaller at high elevation than at low elevation (see Table 2) This was partly due to a smaller albedo bias for stations in the ablation zone than for stations in the accumulation zone".*

**4.** Section 4.8 also seems very short. The authors could provide more discussion of how the albedo differences affect SMB, and how this relates to the biases discussed in other parts of the study.

*A few lines have been added discussing the change in specific SMB when changing the albedo scheme and we refer to the specific SMB figure. Lines 16-21, page 16.*

**Specific Comments:**
1. **P. 1, Line 14:** Suggest changing "specific mass balance" to "specific surface mass balance" for clarity.

   *Has been changed to SMB*

2. **P. 1, Line 16:** Add "through 2014" after "from 1981" to make the time period clear.

   *Has been changed to „for the period 1981-2014" after suggestion from the referee #1*

3. **P. 1, Lines 16-18:** The second part of the sentence doesn't fit with the first part, and contradicts it somewhat. I think the point the authors are trying to make here is that the model can provide a reasonable representation of surface mass balance, but that a major source of uncertainty in this representation is the representation of surface albedo and how it evolves. Please clarify.

   *True, we agree that the second part of the sentence sounds like a contradiction and you are right about your interpretation of the sentence. We have tried to clarify by dividing the sentence and adding some further explanation: Here, we use HIRHAM5 to simulate the evolution of the SMB of Vatnajökull for the period 1981-2014 and show that the model provides a reasonable representation of the SMB for this period. However, a major source of uncertainty in the representation of the SMB is the representation of the albedo, and processes currently not accounted for in RCMs, such as dust storms, are an important source of uncertainty in estimates of snow melt rate.*

4. **P. 2, Line 18:** "Good records" is a bit vague. What is good about them?

We have tried to clarify by changing the sentence to; "Compared to Greenland, observations are recorded in a relatively small area, offering a good opportunity to evaluate the spatial and temporal variability of the HIRHAM5 model on a regional scale"

5. **P. 2, Line 25:** Change "background albedo" to "background bare ice albedo" for clarity.

changed

6. **P. 2, Lines 24-26:** I believe van Angelen et al. (2012) was the first to use this approach. This paper should be cited: van Angelen, J. H., Lenaerts, J. T. M., Lhermitte, S., Fettweis X., Kuipers Munneke, P., van den Broeke, M. R., van Meijgaad, and Smeets, C. J. P. P.: Sensitivity of Greenland Ice Sheet surface mass balance to surface albedo parameterization: a study with a regional climate model, The Cryosphere, 6, 1175-1186, doi: 10.5194/tc-6-1175-2012, 2012.

The paper has been cited and the following sentence has been added; *'This method determining the ice albedo has previously been used by e.g. Angelen et al (2012)'*

7. **P. 3, Line 6:** Note that Brúarjökull and Tungnaárjökull are glaciers that make up part of the Vatnajökull ice cap.

A sentence making this clear has been added

8. **P. 3, Line 25:** How is the summer surface identified?

The summer surface is identified by finding the summer melt layer in snow cores, which is generally easily determined due to a significant amount of dust in the layer.

9. **P. 3, Line 30:** Which MODIS product is used here?

MODIS product MCD43A3 v006 is used. This has been added to the text

10. **P. 4, Line 5:** add "from AWS measurements" after "The turbulent energy fluxes were calculated" for clarity.

Added

11. **P. 6, Lines 20-21:** How is it known that the new particles are generally washed off? Isn't it possible that some of the impurities are scavenged at the surface? (e.g. Doherty et al., 2013) Doherty, S. J., Grenfell, T. C., Forsström, S., Hegg, D. L., Brandt, R. E., and Warren, S. G.: Observed vertical distribution of black carbon and other insoluble light-absorbing particles in melting snow, J. Geophys. Res., 118, 1-17, doi: 10.1002/jgrd.50235, 2013.

This is known from field observations at e.g. Langjökull and Brúarjökull, which has been visited during the summer for the last 20 years. There is a possibility that some of the impurities remain, yes, but most of the particles are washed off and the effect of what might remain is expected to be small compared to the effect of the tephra layers. That this assumption is based on field observations has been clarified in the text.

**12. P. 7, Line 31:** I suggest noting here that the correction was applied so that model results could be compared to AWS measurements at AWS locations.

The sentence was altered to mention this. Now it reads; "The temperature was corrected for the elevation bias in order to compare the model results to the AWS measurements at AWS locations"

**13. P. 8, Lines 14-15:** This repeats some information from section 3.2.3. Since the corrections made in section 3.2.3 are done for the purpose of validation, perhaps the material from section 3.2.3 can be merged into this section.

Section 3.2.3 has been merged with this section (3.2.5).

**14. P. 8, Line 16:** It is unclear what "components" refers to here.

The sentence has been changed so it makes clear that it is the energy balance components

**15. P. 8, Line 22:** Change "temperature, T2m" to "air temperature at 2 m, T2m" for clarity.

changed

**16. P. 9, Line 7:** What is the temperature gradient between?

The atmosphere and the surface. This has been clarified in the sentence.

**17. P. 9, Lines 12-14:** Can the author's elaborate briefly on this? Why are the winds interpolated rather than being calculated within the model?

The wind speeds are interpolated because the lowest atmospheric layer in HIRHAM5 is 10 m. The 2m temperature is interpolated to that height within the model, but the wind speed is not. Therefore we must interpolate it to the AWS height in order to compare it to measurements.

**18. P. 9, Line 31:** Suggest changing "total LW" to "net LW (incoming-outgoing)" radiation

changed

**19. P. 10, Line 10:** By "generally exposed" do the authors mean "every year"?

No, it is not exposed in 2001 and 2011-2013. This has been clarified in the sentence.

**20. P. 10, Line 19:** Since the difficulties in modeling the ELA station have not been elaborated on yet, perhaps the difficulties should be briefly summarized, e.g. "some of the modelling difficulties which affect the ELA station (discussed below), associated with errors in simulating the presence or absence of snow cover…"

Sentence has been deleted due to suggestion from referee #1

21. **P. 11, Line 7:** I believe "underestimating the albedo" should be changed to "overestimating the albedo".

    You're right, it should. It has been changed

22. **P. 12, Line 2:** Suggest changing "total energy was estimated" to "total energy balance was estimated".

    Done

23. **P. 13, Line 1:** "the summer surface was reached" is a bit unclear. Are the authors referring to exposure of bare ice at this location?

    Yes, we are. The sentence has been changed to "bare ice was exposed" for clarity.

24. **P. 13, Line 5:** Change "SW radiation" to "net SW radiation" for clarity.

    Changed

25. **P. 13, Line 8:** Again "net SW radiation" would be clearer.

    Changed

26. **P. 16, Line 4:** Be more clear about what is underestimated.

    Done. We have added that the underestimation is of the energy balance components.

27. **P. 16, Line 12:** It is known that the model simulates surface temperatures well, as discussed in the previous paragraph. Perhaps it is better to say that the accuracy of outgoing longwave radiation is consistent with the ability of the model to capture surface temperatures.

    The sentence has been changed to reflect this. It now reads; "Whereas the modelled outgoing LW radiation component is within the uncertainty of the LW observations at the five stations, *which is consistent with the ability of the model to capture surface temperatures*, there was a larger difference between the modelled and measured outgoing SW radiation"

28. **P. 16, Lines 15-16:** The better agreement with observations as compared with a fixed albedo, though obvious given the wide spread of observed values, is not mentioned in the results section. If mentioned here, it should also be mentioned in Section 4.3.

    You're right, it wasn't. That part of the sentence has been deleted.

29. **P. 16, Lines 30-31:** This sentence is confusing. It makes it seem as if the average modeled mass balance for 1981-2014 is being compared with the average for 1995-2014 from observations.

Rather, the model results for 1995-2014 were compared with observations for 1995-2014. Please clarify.

*You're right. We have tried to clarify this by changing the sentence to;* *'The mean specific summer, winter and net mass balances are reconstructed for all of Vatnajökull from 1981-2014, and estimates of the specific SMB based on in situ SMB measurements are compared to the reconstructed specific SMB for the period 1995-2014.'*

30. **Table 2:** In the caption, the meaning of the parameters in column 1 should be explained, as is done for Table 3.

    *The meaning of the parameters has been added to the beginning of the caption*

31. **Figure 1:** The weather station names are not consistent with the names in the text. For example "B-abl" should be "B-AB" to be consistent with the text. Also, in the caption, it should be pointed out that the unlabeled sites in Fig. 1a were not used in the study. Optionally, the symbols could be a different color to emphasize this. Perhaps Brúarjökull and Tungnaárjökull could also be labeled on the map for clarity. The lines on Fig. 1b are not explained. I suppose these connect mass balance sites collected along a transect. Finally, the labels (a) and (b) should be added for the sub-plots.

    *The names of the stations have been changed in the figure, and we point out in the label that only labeled AWSs are used in this study. A description of the colored lines has also been added (they do connect mass balance sites collected along a transect), and the labels (a) and (b) has been added to the plots.*

32. **Figure 10:** (Caption) Add the years of the observational period for clarity.

    *Added*

**Technical Corrections:**
1. **P. 1, Line 3:** Suggest changing "describes the albedo with an exponential decay with time…" to "allows albedo to exponentially decay with time…"
2. **P. 4, Line 1:** Change "lat/lon" to "latitude-longitude coordinates"
3. **P. 5, Line 28:** Change "The found best-fit values were…" to "The best-fit values were found to be…"
4. **P. 5, Line 30:** Change "Refreshment of albedo to the maximum value only occurs…" to "Albedo is only refreshed to the maximum value if.."

    *Changed! Thanks*

5. **P. 6, Line 11:** I believe there is a typo in the equation. Should "dn+1" be "dt+1"?

    *You're right, it should be. It has been changed.*

6. **P. 6, Line 23:** Change "How much this" to "The extent to which"

Changed

7.  **P. 7, Line 4:** Change "The model is here run" to "For this study, the model is run"

    The sentence has been deleted and the period added to the first line after suggestion from referee #1

8.  **P. 7, Line 10:** Change "allows a quick and thorough" to "allows for a quick and thorough"
9.  **P. 8, Line 5:** Change "like for example that of the albedo" to "including, for example, the albedo parameterization,"
10. **P. 8, Line 31:** Change "with 0.8 K overall" to "by 0.8 K on average"

    Done

11. **P. 9, Line 1:** Change "but with less than 0.6 K" to "by less than 0.6 K"; change "it for example" to "for example, it"
12. **P. 9, Line 20:** Change "larger errors-" to "larger errors;"
13. **P. 10, Line 12:** Change "down to 0.03" to "as low as 0.03"; Change "the total overestimation" to "the average overestimation"
14. **P. 10, Line 27:** Change "that low in situ…" to "that a low in situ…".
15. **P. 11, Line 28:** Change "comparisons statistics" to "comparison statistics".
16. **P. 12, Lines 10-12:** This sentence is rather long. I suggest splitting it into two sentences.
17. **P. 14, Line 16:** Change "one-third that of the AWS sites…" to "one-third the difference with respect to the AWS sites…"

    Done

18. **P. 14, Line 27:** Change "back to 1981" to "extending back to 1981".

    Changed to "for 1981-2014"

19. **P. 16, Line 6:** Change "comparisons only uses" to "comparisons only use".
20. **P. 16, Line 12:** Change "there was a larger differences" to "there was a larger difference".
21. **P. 16, Line 19:** Change "and that the model does not account" to "and the fact that the model does not account"
22. **P. 16, Line 20:** Change "way to include" to "means of capturing"
23. **P. 17, Line 21:** Change "could be including a stochastic…" to "could be to include a stochastic…"
24. **P. 17, Line 31:** Change "with 0.06 m" to "by 0.06 m".
25. **P. 18, Line 13:** Change "like for example ERA-20C" to "for example, with the ERA20C reanalysis".

    Done

26. **Figure 3:** The axis for Fig. 3b is a bit confusing. I suggest removing the 100, and leaving 0 for all plots.

Changed

27. **Figure 4:** suggest adding "from AWS stations" after "fluxes calculated" for clarity.
28. **Figure 6:** The caption seems to be erroneously in italics.
29. **Figure 11:** Change "used AWS" to "AWS stations used in this study"

Done! thanks